# Does Math Reasoning Improve General LLM Capabilities? Understanding Transferability of LLM Reasoning

**Maggie Ziyu Huan** [* 1 2]  **Yuetai Li** [* 3]  **Tianyu Zheng** [* 4]  **Xiaoyu Xu** [5]  **Seungone Kim** [1]  **Minxin Du** [5]
**Radha Poovendran** [3]  **Graham Neubig** [1]  **Xiang Yue** [1]

## Abstract

Math reasoning has become the poster child of progress in large language models (LLMs), with new models rapidly surpassing human-level performance on benchmarks like MATH and AIME. But as math leaderboards improve week by week, it is worth asking: *do these gains reflect broader problem-solving ability or just narrow overfitting?* To answer this question, we evaluate over 20 open-weight reasoning-tuned models across a broad suite of tasks, including math, scientific QA, agent planning, coding, and standard instruction-following. We surprisingly find that most models that succeed in math fail to transfer their gains to other domains. To rigorously study this phenomenon, we conduct controlled experiments using math-only data with two widely-used methods: Reinforcement Learning (RL) and Supervised Finetuning (SFT) with detailed ablations. On top of the observation that RL-tuned models transfer better than SFT-tuned models, we identify on-policy fine-tuning as the key mechanism underlying cross-domain transfer, regardless of whether the training signal comes from RL or supervised learning. Latent-space representation and token-space distribution shift analyses reveal that off-policy SFT induces substantial representation and output drift, while on-policy RL preserves general-domain structure. Our results suggest a need to rethink the post-training recipes, particularly the reliance on off-policy SFT-distilled data to advance reasoning models.

---

[1]Language Technology Institute, Carnegie Mellon University [2]Department of Computer and Information Science, University of Pennsylvania [3]Department of ECE, University of Washington [4]M-A-P [5]Department of EEE, The Hong Kong Polytechnic University. Correspondence to: Maggie Huan <ziyuh@upenn.edu>, Xiang Yue <xyue2@andrew.cmu.edu>.

*Proceedings of the 43rd International Conference on Machine Learning*, Seoul, South Korea. PMLR 306, 2026. Copyright 2026 by the author(s).

## 1. Introduction

Over the past years, the community has raced to push large language models (LLMs) to new heights on math-centric reasoning benchmarks such as MATH (Hendrycks et al., 2021b) and AIME. A steady stream of *reasoning-tuned* models (Muennighoff et al., 2025; Guha et al., 2025) now advances the state of the art on math benchmarks nearly every week, with some even surpassing the average performance of human experts (Team, 2025a; OpenAI, 2024).

The appealing performance on math reasoning is understandable: problems are well-posed, solutions are unambiguous, and evaluation is easily verifiable, often just a single number or expression (Luo et al., 2025). This clarity has made math a popular proxy task of LLM reasoning, and researchers have developed increasingly sophisticated training recipes to maximize model performance on mathematical reasoning (Wang et al., 2024a; Yue et al., 2024a; Luo et al., 2023; Shao et al., 2024; Wei et al., 2023). This trend, on one hand, should be encouraged. Mathematics is often considered the foundational language of science, and enabling machines to reason precisely over math is central to the long-term vision of automated scientific discovery (Mishra et al., 2022; Jones et al., 2026a;b; Yifei et al., 2026). On the other hand, real-world tasks extend far beyond math. The majority of user-facing applications, question answering, dialogue, instruction following, require broader linguistic and commonsense competence that math alone does not test (Ma et al., 2025; Bai et al., 2025; Hu et al., 2026).

This raises a natural question: *Do improved math reasoning abilities transfer to general LLM capabilities?* Specifically, can gains in solving math problems transfer to other reasoning domains (e.g., scientific QA (Welbl et al., 2017), coding (Jain et al., 2025), agent planning (Xie et al., 2024), logical deduction (Dziri et al., 2024)) and to tasks (e.g., conversational QA (Reddy et al., 2019), instruction following (Zhou et al., 2023)) that do not require extensive reasoning? To investigate, we evaluate over 20 representative open-weight reasoning models, all of which exhibit impressive performance on recent math benchmarks across a suite of other reasoning and non-reasoning tasks. To quantitatively analyze this problem, we propose *Transferability Index*, a

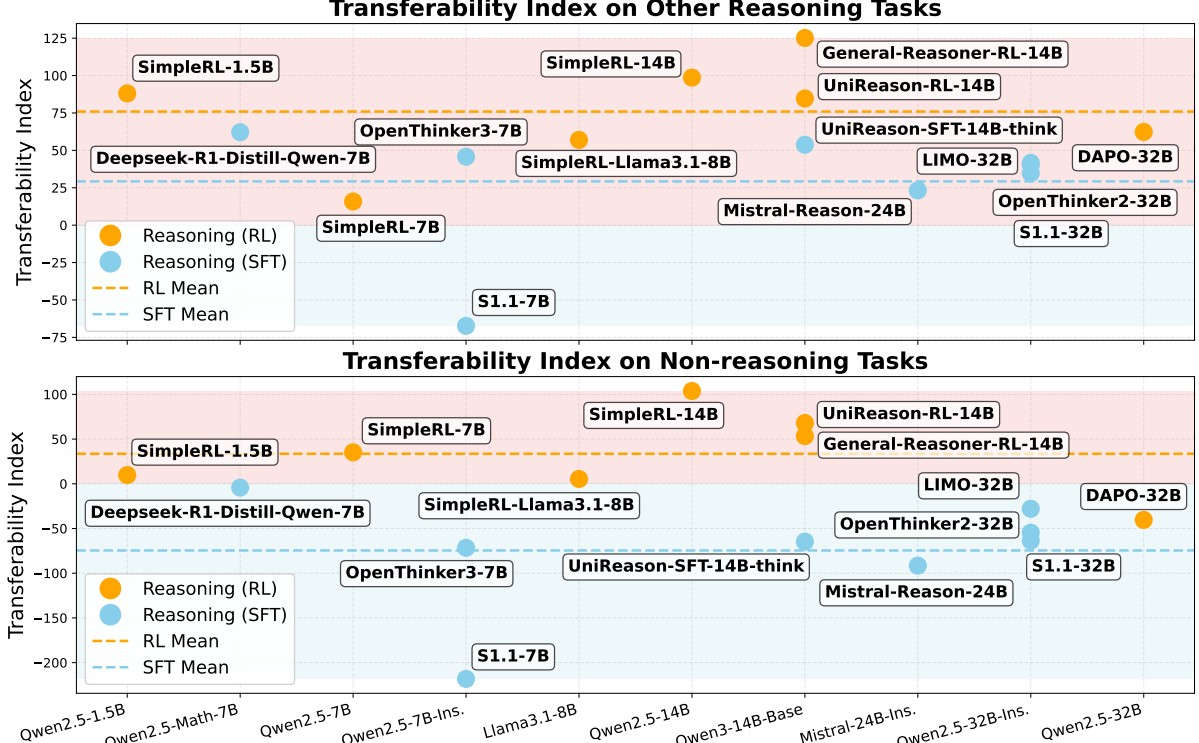

*Figure 1.* Transferability of mathematical reasoning to other reasoning and non-reasoning tasks. The Transferability Index measures a model's ability to transfer performance from mathematics to other domains, with positive values indicating successful transfer and negative values indicating performance degradation. Details of this metric can be found in Section 2.1. RL models consistently outperform SFT models, regardless of model size, architecture, or training data, demonstrating superior transferability.

metric to measure how reasoning models can transfer their capabilities from one domain to another. Surprisingly, as shown in Figure 1, we find that some of these models *fail* to transfer their improved mathematical reasoning capabilities to other domains while others succeed.

What drives this divergence? Model recipes vary widely in size, data distribution, and architecture. Yet among various parts, we identify one factor that consistently predicts transferability: the fine-tuning *paradigm*. Across families and sizes, models fine-tuned using on-policy reinforcement learning (RL) (Su et al., 2025; Yeo et al., 2025) exhibit much stronger performance in non-math tasks than those trained with off-line supervised fine-tuning (SFT) (Yue et al., 2024a;b), which often show signs of catastrophic forgetting over a wide range of non-math tasks.

To validate this observation, we conduct a controlled study using most commonly used training paradigms: on-policy RL and off-line SFT in tuning reasoning models. We fine-tune Qwen3-14B (Team, 2025b) on the high-quality math dataset derived from MATH and DeepScaler (Luo et al., 2025). For SFT, we construct targets via rejection sampling using Qwen3-32B, keeping only teacher responses that yield correct final answers. For RL, we apply a standard GRPO (Shao et al., 2024) setup using answer correctness as the reward. As shown in Table 1, the results mirror our large-scale audit: on-policy RL-tuned models transfer well to non-math domains, despite being trained solely on math queries, while off-line SFT-tuned models do not. We validated this finding with more ablations with broader settings, which could be found in the Appendix A.1.

To better understand why this occurs, we use two diagnostic tools: (1) latent-space principal component analysis (PCA) on hidden states across layers, and (2) KL-divergence on token distributions before and after fine-tuning. These methods allow us to quantify how much the model's internal representations and output space shift during training. We find that off-line SFT induces significant drift in both latent and output spaces, especially for non-reasoning inputs, whereas on-policy RL better preserves the geometry of internal features and the stability of the token distributions.

Building on these observations, we refine our understanding of which concrete mechanisms drive the differences between the widely used SFT and RL. Specifically, we conduct an ablation study to learn which components contribute to the significant variances in transferability. Results show that the sampling distribution, credit assignment, and negative gradient contribute more, while KL regularization plays only a subtle role in the transferability of LLM reasoning.

## 2. Phenomena: Performance Discrepancies of Reasoning Models

### 2.1. Observed Performance Discrepancies Between RL and SFT

**Setup**. We evaluate over 20 off-the-shelf reasoning models on different benchmarks. Namely, we select benchmarks from the following three categories: (1) **math reasoning tasks**: MATH500 (Hendrycks et al., 2021b), AIME24, AIME25, OlympiadBench (He et al., 2024), which contain mathematical problems only; (2) **other reasoning tasks**: LiveCodeBench (Jain et al., 2025), GPQA-Diamond (Rein et al., 2024), ACPBench (Kokel et al., 2025), HeadQA (Vilares & Gómez-Rodríguez, 2019), which contain more general reasoning questions, such as medical reasoning, code generation, and language-based agent planning tasks; (3) **non-reasoning tasks**: CoQA (Reddy et al., 2019), IFEval (Zhou et al., 2023), HaluEval (Li et al., 2023), MC-TACO (Zhou et al., 2019), which contain factual, alignment, or conversational problems such as commonsense question answering and instruction-following. We used *accuracy* to evaluate the models' performance. Detailed explanation about experiment setup, benchmarks, and evaluation metrics can be found in Appendix A.5. To better evaluate the model's transferability across a wide range of task groups, we define **Transferability Index** (TI) as follows:

*(1) Per-benchmark gain and normalization. $z$-normalization* makes gains comparable within each group. For each group $g \in \{\texttt{math}, \texttt{other}, \texttt{non}\}$ and benchmark $b \in \mathcal{B}_g$,

$$\Delta R_b = R_b^{\text{model}} - R_b^{\text{base}}, \sigma_g = \text{Std}\{\Delta R_b : b \in \mathcal{B}_g\},$$

$$\delta_b = \frac{\Delta R_b}{\sigma_g}.$$

*(2) Robust gain score and difficulty weighting.* We then compute the domain index to normalize benchmark differences. We control extremes with a signed square-root and up-weight harder tasks, and used a group-level *Domain Index* aggregates these weighted, robust gains:

$$s_b = \text{sign}(\delta_b) |\delta_b|^{1/2}, \hat{w}_b = \frac{100 - R_b^{\text{base}}}{\sum_{u \in \mathcal{B}_g} w_u},$$

$$\text{DI}_g = \sum_{b \in \mathcal{B}_g} \hat{w}_b s_b.$$

*(3) Transferability relative to math.* Finally, TI is the ratio of the group's improvement to the math group's improvement:

$$\text{TI}_g(\%) = \frac{\text{DI}_g}{\text{DI}_{\texttt{math}}} \times 100, g \in \{\texttt{other}, \texttt{non}\}.$$

A positive $\text{TI}_g$ indicates positive transfer in group $g$, and higher $\text{TI}_g$ indicates the finetuned model achieves stronger

and more consistent improvements in transferring math capabilities to other domains. To figure out the effect of introducing signed square-root into TI, we conducted an ablation study in Appendix A.2.

**Results.** Figure 1 and Table 11 compare transferability indices across models. Transferability varies with finetuning method, model size, and architecture; the dominant factor is the *method*: RL-tuned models consistently achieve higher indices on both other-reasoning and non-reasoning tasks, whereas SFT-trained models often yield negative $\text{TI}_{\text{non}}$. This suggests that on-policy RL reinforces reasoning while minimally perturbing general-domain representations, whereas off-policy SFT on large static reasoning corpora can overspecialize the latent space, degrading non-reasoning performance.

### 2.2. Control Study

Motivated by our findings in Section 2.1, we design a lightweight controlled study to directly compare two mostly used paradigms for training reasoning models: off-policy SFT and on-policy RL on an identical dataset. Concretely, we start from a small, high-quality mathematics dataset (see Appendix A.5.2 for details), then query a strong teacher model (Qwen3-32B-Instruct) to extract complete chain-of-thought (CoT) reasoning traces with reject sampling. These CoT traces become our SFT training targets, while the original answer labels serve as the rewards for RL. This alignment ensures both paradigms learn from the same data samples. Then, we take the Qwen3-14B-Base model and fine-tune it in two ways: (i) *SFT* on the teacher-generated CoT traces; (ii) *RL* using only the groundtruth. We name our model **UniReason**. We compare against the Qwen3-14B-Base model. Evaluation is conducted on three benchmark groups mentioned above using *accuracy*. Details about training datasets, baseline models, and hyperparameters could also be found in Appendix A.5. Our experimental results on three groups of benchmarks (see Table 1) reveal a consistent pattern:

- On **math reasoning**, our UniReason-Qwen3-14B(RL) model climbs to 55.7% on AIME24, 87.8% on MATH500, and 33.8% on OlympiadBench, outperforming corresponding SFT-based models.

- For **other reasoning** tasks, SFT-based models make uneven progress (e.g. UniReason-Qwen3-14B-SFT-*think* scores 55.9% on GPQA), whereas RL fine-tuning yields significant lifts: UniReason-Qwen3-14B(RL) gains 1.8% on GPQA, and 17.1% on LiveCodeBench2 over SFT.

- Crucially, in **non-reasoning** evaluations, SFT models stagnate or decline, while the RL model recovers and exceeds the base in nearly all the benchmarks.

*Table 1.* Performance of our model and baseline models on math reasoning, other reasoning and non-reasoning tasks. UniReason-Qwen3-14B-*think* (SFT) denotes the model is trained with the dataset distilled from Qwen3-32B thinking mode, UniReason-Qwen3-14B-*no-think* (SFT) denotes it is trained with Qwen3-32B non-thinking mode. The grey-colored row represents the performance difference between RL and the highest SFT score. Additional studies on Qwen3-4B, Llama3.1-8B models; training with more diverse training data and on more evaluation benchmarks could be found in Appendix A.1.

| Math Reasoning | | | | | | |
|---|---|---|---|---|---|---|
| **Model** | **AIME24** | **AIME25** | **MATH500** | **Olympiad** | **Average** | |
| Qwen3-14B-Base | 13.0 | 9.3 | 60.4 | 27.9 | 27.7 | - |
| UniReason-Qwen3-14B-*think* (SFT) | 52.0 | 37.0 | 85.0 | 25.0 | 49.8 | - |
| UniReason-Qwen3-14B-*no-think* (SFT) | 16.0 | 13.0 | 77.2 | 22.7 | 32.3 | - |
| UniReason-Qwen3-14B (RL) | 55.7 | 38.0 | 87.8 | 33.8 | 53.8 | - |
| Δ (RL - SFT) | +3.7 | +1.0 | +2.8 | +8.8 | +4.1 | - |
| **Other Reasoning** | | | | | | |
| **Model** | **GPQA** | **LiveCodeBench2** | **ACPBench** | **HeadQA** | **Average** | $TI_{other}$ |
| Qwen3-14B-Base | 42.6 | 29.7 | 10.7 | 37.6 | 30.2 | - |
| UniReason-Qwen3-14B-*think* (SFT) | 55.9 | 21.9 | 68.6 | 34.8 | 45.3 | +52.2 |
| UniReason-Qwen3-14B-*no-think* (SFT) | 48.7 | 23.5 | 69.3 | 35.0 | 44.1 | +165.4 |
| UniReason-Qwen3-14B (RL) | 57.7 | 40.6 | 65.4 | 40.2 | 51.0 | +82.3 |
| Δ (RL - SFT) | +1.8 | +17.1 | -3.9 | +5.2 | +5.7 | - |
| **Non-Reasoning** | | | | | | |
| **Model** | **CoQA** | **MC-TACO** | **IFEval** | **HaluEval** | **Average** | $TI_{non}$ |
| Qwen3-14B-Base | 10.0 | 67.7 | 69.2 | 35.7 | 45.7 | - |
| UniReason-Qwen3-14B-*think* (SFT) | 1.7 | 38.2 | 42.3 | 2.3 | 21.1 | -104.1 |
| UniReason-Qwen3-14B-*no-think* (SFT) | 5.3 | 66.1 | 41.4 | 3.3 | 29.0 | -278.9 |
| UniReason-Qwen3-14B (RL) | 28.2 | 74.0 | 70.0 | 40.7 | 53.2 | +52.2 |
| Δ (RL - SFT) | +22.9 | +7.9 | +27.7 | +37.4 | +24.0 | - |

- From Table 1, our RL-tuned model achieves positive $TI_{other}$ and $TI_{non}$, meaning its math improvements also transfer to other reasoning and even boost non-reasoning performance. In contrast, SFT-based models often have negative $TI_{non}$, indicating poor transfer beyond reasoning tasks.

These results show that RL-tuned reasoning models perform generally better than SFT-based models on both reasoning and non-reasoning tasks when carefully controlling other factors. Especially, our UniReason model is trained on a single distilled math dataset, but it still preserves and even improves general-domain performance while showing strong reasoning gains. To further validate our findings under different model sizes, learning rates and architectures, we applied this setting to Qwen3-4B model and Llama3.1-8B model, the results could be found in Appendix A.1. We also evaluated our model on several more benchmarks. The results show a consistent pattern with our control study.

## 3. Latent Representation Shifts: Insights from PCA Analysis

As discussed in Section 2.1, applying SFT to the Qwen model improves reasoning abilities such as mathematical

problem-solving and code generation, but impairs general-domain performance. We observe that most SFT models fail to transfer their improved mathematical reasoning capabilities to other domains. In contrast, our controlled study shows that RL models transfer well to non-math domains, despite being trained solely on math queries, whereas SFT models do not.

To understand the underlying cause of this transferability gap, we employ PCA shift analysis to examine how the internal feature geometry of the model evolves under different training paradigms, model sizes, and model families across diverse query distributions. Recent studies (Xu et al., 2025a; Zheng et al., 2025) demonstrate that PCA shift analysis provides a sensitive and interpretable measure of representational changes relevant to task performance. Importantly, changes in model parameters do not always correspond to functional differences: large weight updates may leave outputs unchanged, while subtle parameter modifications can lead to significant shifts in the activation distribution. By focusing on hidden representations, PCA shift directly captures how the model encodes and processes information, offering a more faithful account of its internal knowledge state than parameter-based metrics. This perspective allows us to distinguish between true knowledge erasure and parameterization changes that leave the underlying feature space intact.

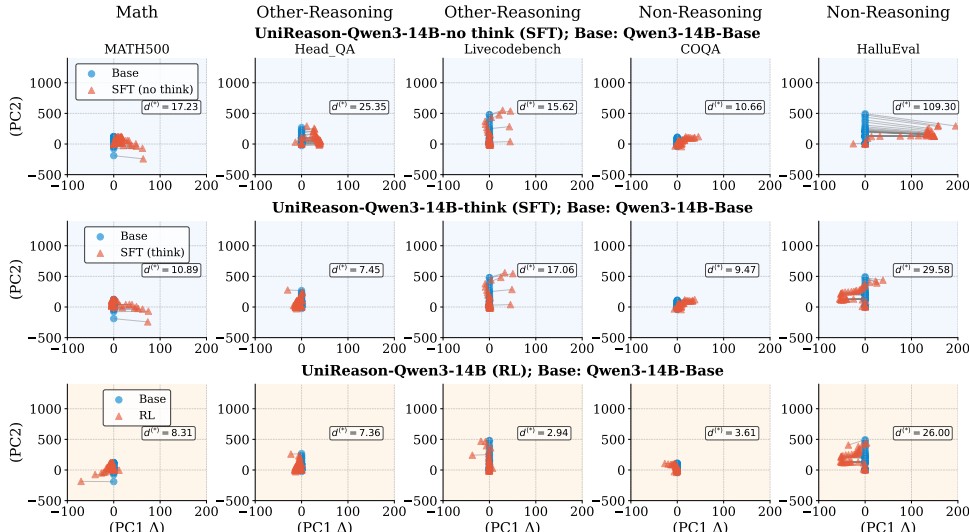

*Figure 2.* PCA shift of Qwen3-14B-Base across different training methods and tasks. $d^{(*)}$ is the Euclidean distance between representation centroids before and after training. The first two rows show models trained with SFT, and the last row shows models trained with RL. RL training results in the smallest PCA shift for all task types, suggesting more stable latent representations.

Furthermore, since transferability fundamentally relies on the alignment and stability of learned representations across tasks or domains, PCA shift is particularly effective for diagnosing changes that may impact cross-domain performance. Shifts in principal components reveal whether the model's internal feature space remains suitable for knowledge transfer or has been disrupted by post-training interventions.

### 3.1. Experimental Setup

**Models and Tasks.** In Section 2.1, we observe that models trained on math datasets show moderate transferability on other reasoning tasks. We perform PCA shift analyses on the corresponding models and tasks, aiming to critically assess these phenomena from a feature-space perspective.

**Evaluation.** Given input queries $\mathcal{X}$, we extract hidden states $\mathbf{H}_i^{(*)}$ at each layer $i$ for each model state $(*) \in \{orig, updated\}$. Applying PCA ($n = 2$) to $\mathbf{H}_i^{(*)}$, we compute the mean projection $m_{i,1}^{(*)}$ onto the first principal direction (PC1) and $m_{i,2}^{(*)}$ onto the second (PC2). The PCA shift is defined as $\Delta m_{i,1}^{(*)} = m_{i,1}^{(*)} - m_{i,1}^{orig}$ for PC1, while for PC2, we directly report $m_{i,2}^{(*)}$ as an indicator of change.

### 3.2. Investigating Latent Space Shift

To quantify the overall latent shift, we define a *representation center* for each model state $(*)$ as the mean of PCA-projected coordinates across all layers: $\mathbf{z}^{(*)} = \frac{1}{L} \sum_{i=1}^{L} \mathbf{z}_i^{(*)}$, where $L$ denotes the total number of layers

*Table 2.* Mean PCA shift magnitudes on math, reasoning, and non-reasoning tasks for Qwen3-14B and our finetuned models. RL models exhibit lower shifts than SFT models. **Other** denotes Other Reasoning tasks, **Non** denotes non-reasoning tasks.

| Model | Math | Other | Non |
|---|---|---|---|
| Qwen3-14B-*no-think* | 40.4 | 13.9 | 129.8 |
| Qwen3-14B-*think* | 76.5 | 38.8 | 152.0 |
| UniReason-Qwen3-14B-*no-think* (SFT) | 21.4 | 10.9 | 113.7 |
| UniReason-Qwen3-14B-*think* (SFT) | 19.2 | 6.7 | 38.2 |
| UniReason-Qwen3-14B (RL) | 8.5 | 3.5 | 36.9 |

and $\mathbf{z}_i^{(*)} = (\Delta m_{i,1}^{(*)}, m_{i,2}^{(*)})$ is the vector of PCA coordinates for layer $i$ in state $(*)$. The latent shift between two model states, such as the original (base) and an updated model, is then measured by the Euclidean distance: $d^{(*)} = \left\| \mathbf{z}^{(*)} - \mathbf{z}^{(orig)} \right\|_2$.

Based on the analyses in Appendix A.6, RL-based training proves essential for developing language models that maintain a strong balance between general-domain and reasoning capabilities. Motivated by this, we further analyze our proposed models in the controlled study. As shown in Table 2, RL models achieve the lowest PCA shift magnitudes across math, other-reasoning, and non-reasoning tasks. Figure 2 further supports these findings, illustrating that the RL model consistently yields minimal and tightly clustered latent shifts across diverse benchmarks. In contrast, SFT models, particularly those without explicit reasoning signals, exhibit more scattered shifts. These results and the evaluations in Section 2.1, highlight the advantage of RL over SFT. They underscore the importance of a well-balanced

optimization objective rather than isolated interventions, for mitigating catastrophic forgetting while preserving performance in LLMs.

## 4. Token Distribution Shifts: Insights from KL Divergence and Rank Analyses

In this section, we conduct token-level analyses to further examine the distribution shift of RL and SFT models trained on mathematical reasoning data.

### 4.1. Setup

KL-divergence serves as a standard metric for measuring differences between probability distributions. For token rank shift analysis, we first generate tokens using the fine-tuned model, then decode these same tokens using the backbone model to determine their original ranking positions. The rank shift is calculated as the difference in token rankings between the fine-tuned model and the backbone model for each token (Li et al., 2025c; Lin et al., 2023). Following the observations in Section 2.1, we perform additional token-distribution analyses on the corresponding models and tasks to assess the model distribution shift from a token-space perspective. Specifically, we employ KL-divergence and token rank shift metrics to analyze distribution shifts between models.

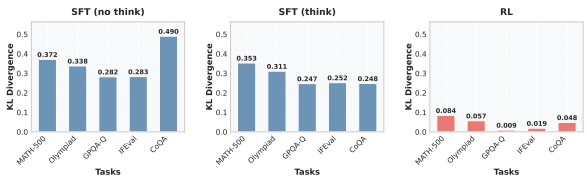

*Figure 3.* KL divergence analysis of RL and SFT models. Higher KL divergence indicates greater distribution shifts from the original backbone model. We observe that RL models consistently exhibit lower KL divergence compared to SFT models across different tasks, suggesting less distribution shift during training.

### 4.2. Result

**RL models exhibit lower KL-divergence from backbone models.** In Figure 3, we observe that the KL divergence of SFT models on both reasoning and non-reasoning tasks is significantly larger than that of RL models. This indicates that RL models exhibit substantially less distribution shift from the token distribution level compared to SFT models. For instance, UniReason-Qwen3-14B-SFT-*no-think* demonstrates KL divergences of 0.372 and 0.283 on MATH-500 and IFEval respectively compared to the backbone model, whereas UniReason-Qwen3-14B(RL) achieves considerably lower KL divergences of only 0.084 and 0.019 on the corresponding tasks.

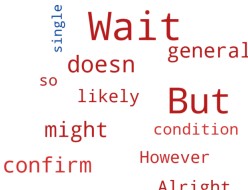 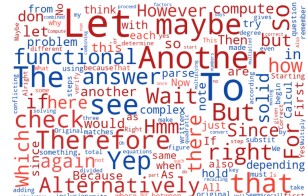

*Figure 4.* Word clouds showing significantly shifted tokens between UniReason-Qwen3-14B-RL (left) and UniReason-Qwen3-14B-SFT-*think* (right). Tokens are extracted based on frequency and rank shifts compared with base model then categorized as logical-structural words (in red) or content-specific words (in blue). The RL model promptly shifts logic-related tokens such as *But* and *So* while the SFT model shifts including many irrelevant tokens.

**RL models demonstrate reduced token rank shifts.** Across both reasoning and non-reasoning tasks, RL models show markedly smaller average token-rank shifts than SFT models as shown in Figure 16 in Appendix A.8. For example, UniReason-Qwen3-14B (RL) averages only 0.98 positions of shift, whereas the SFT no-think variant reaches 10.6. This indicates RL tends to preserve the base token distribution while SFT induces larger reordering. Detailed position-wise trajectories are provided in Figure 19 in Appendix A.8.

**RL models selectively shift task-relevant tokens, while SFT models shift numerous irrelevant tokens.** A case study in Table 14 shows RL moving a small set of task-relevant tokens (e.g., "define", "add", "second", "number" for reasoning; "<<", ">>", "write", "formally" for non-reasoning), while SFT shifts far more tokens (e.g., 390 for reasoning / 158 for non-reasoning), including many that are query-irrelevant and often inject reasoning markers into non-reasoning prompts. The token-frequency word cloud in Figure 4 further visualizes this selectivity gap. More detailed information could be found in Appendix A.8.

## 5. Ablation: Which Components of RL Drive Transferability?

Earlier sections observed that reinforcement-style fine-tuning transfers more effectively than standard supervised fine-tuning. Latent and output space analyses revealed that RL updates produce small, targeted "marginal shifts" around the base model, whereas SFT often causes substantial drift. This section sharpens the inquiry: *Which specific components of RL updates contribute to improved transferability?*

We contrast SFT and RL by aligning their objectives and gradients, yielding a *unified surrogate loss* that makes their differences explicit along four levers: (i) **sampling distribution**: SFT trains *off-policy* (offline) on a fixed dataset, whereas RL samples *on-policy* from the live model; (ii) **credit assignment**: uniform weighting versus advantage-

*Table 3.* Ablation settings and their corresponding sampling distributions, credit weights, and KL-regularization coefficients. The two shaded rows mark the *standard baselines*: Off-policy SFT and On-policy RL.

| Setting | Sampling $q$ | Weights $w$ | KL Reg. $\beta$ |
|---|---|---|---|
| Off-policy SFT | $\delta_{y=y^\star}$ | 1 | 0 |
| On-policy SFT | $\pi_\theta$ | 1 (reject sample) | 0 |
| Off-policy RL | $\delta_{y=y^\star}$ | Advantage $A_t$ | 0 |
| On-policy RL (no KL) | $\pi_\theta$ | Advantage $A_t$ | 0 |
| On-policy RL | $\pi_\theta$ | Advantage $A_t$ | $> 0$ |

*Table 4.* Performance of all the ablation models. Scores are average scores on math reasoning, other reasoning and non-reasoning tasks. **Other** denotes Other Reasoning tasks, **Non** denotes non-reasoning tasks.

| Model | Math | Other | Non | $\mathrm{TI_{other}}$ | $\mathrm{TI_{non}}$ |
|---|---|---|---|---|---|
| Qwen3-8B-Base | 27.6 | 23.6 | 33.6 | - | - |
| Qwen3-8B-Off-policy-SFT | 41.9 | 34.4 | 26.6 | 18.3 | -40.5 |
| Qwen3-8B-On-policy-SFT | 33.7 | 35.7 | 35.0 | 68.6 | 30.2 |
| Qwen3-8B-Off-policy-RL | 45.5 | 35.9 | 31.7 | 36.4 | 4.5 |
| Qwen3-8B-On-policy-RL (no KL) | 37.1 | 38.2 | 35.8 | 65.6 | 39.3 |
| Qwen3-8B-On-policy-RL | 38.6 | 39.9 | 35.0 | 63.7 | 32.4 |

weighted updates that emphasize task-salient tokens; (iii) **KL regularization**: penalizing deviations from a reference policy such as PPO or GRPO; (iv) **negative gradient**: When sampling from $\pi$, RL learns from negative examples, while SFT only imitates positive ones.

## 5.1. Contrasting Objectives

For prompts $x$ and completions $y$, let $\pi_\theta(y \mid x)$ denote the current policy, and $\pi_{\mathrm{ref}}(y \mid x)$ denote a fixed reference policy (e.g., the initialization).

**Supervised Fine-tuning.** With reference completions $y^\star$, the objective is:

$$\mathcal{L}_{\mathrm{SFT}}(\theta) = -\mathbb{E}_{(x,y^\star)\sim\mathcal{D}}\big[\log \pi_\theta(y^\star \mid x)\big]. \quad (1)$$

**Reinforcement Learning.** For the same prompts, we sample $y \sim \pi_\theta$ and weight each sample by an *advantage* $A(x, y)$:

$$\mathcal{L}_{\mathrm{RL}}(\theta) = -\mathbb{E}_{x\sim\mathcal{D}}\mathbb{E}_{y\sim\pi_\theta(\cdot\mid x)}\big[A(x,y) \cdot \log \pi_\theta(y \mid x)\big] \quad (2)$$

Usually, a KL term is added to prevent the policy model from being too far from the initialized model. We generalize these objectives using:

$$\begin{aligned}\mathcal{L}_{q,w,\beta}(\theta) = &-\mathbb{E}_{x\sim\mathcal{D}}\mathbb{E}_{y\sim q(\cdot\mid x)}\big[w(x,y) \cdot \log \pi_\theta(y \mid x)\big] \\ &+ \beta\,\mathbb{E}_{x\sim\mathcal{D}}\big[\mathrm{KL}(\pi_\theta(\cdot \mid x) \,\|\, \pi_{\mathrm{ref}}(\cdot \mid x))\big]\end{aligned} \quad (3)$$

## 5.2. Ablation Results

**Setup.** For ablation study, we used 5 settings from Off-policy and On-policy SFT, Off-policy and On-policy RL with and without KL. We used the same set of math training queries and a Qwen3-8B-Base model. For on-policy SFT: we sample a batch of training queries and let the policy model generate the responses as in standard RL training. For off-policy RL, we use the Qwen3-32B model in think mode to generate $n = 8$ responses for each query. We utilized GRPO's objective with KL divergence and used the response with the highest reward for gradient computation. For Off-policy SFT, we used data generated from Qwen3-32B model

in think mode with reject sampling. On-policy RL used the same math dataset as our main models. We compare RL against SFT after matching the number of rollouts (n=8) and, within each setting, the sampling distribution: on-policy RL and on-policy SFT both train on policy-generated rollouts; off-policy RL and off-policy SFT both use Qwen3-32B rollouts. The results are in Table 4. We also found that adding L2-SP regularization would not help SFT models transfer to other tasks under Qwen3-8B models as shown in Table 6.

**Sampling distribution is critical**. The results in Table 4 show that on-policy methods outperform off-policy methods across both evaluation categories and training paradigms. On-policy SFT achieves higher average scores on Other Reasoning tasks and Non-Reasoning tasks compared to off-policy SFT. A similar trend is observed when KL regularization is applied to RL: on-policy RL continues to yield superior transferability on Other Reasoning and Non-Reasoning tasks. These results demonstrate that on-policy methods consistently transfer better than off-policy methods. One possible explanation is that on-policy training exposes models to data generated under their own evolving policy, allowing updates to directly align with the distribution the model will face at inference. This reduces distribution mismatch and mitigates overfitting to fixed samples, leading to stronger generalization. Gradient Norm during training also shows that on-policy methods update the gradient in small and conservative steps compared to off-policy methods, which aggressively update their gradient especially from the start. Details can be found in Figure 5 and Appendix A.7.

**Credit assignments and negative examples matter**. Experiments on credit assignment and negative gradients show that adding these mechanisms not only improves transferability but also increases response length during training. In off-policy settings, RL achieves higher average scores on both Other Reasoning and Non-Reasoning tasks. Under on-policy training, RL outperforms SFT on Math and Other Reasoning tasks, while showing similar performance on Non-Reasoning tasks. Our ablation therefore shows that these two signals jointly improve transferability; because rejection sampling removes both at once, the design does

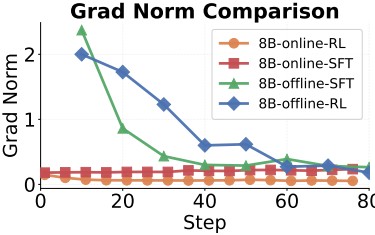

*Figure 5.* Grad norm comparison across on-policy and off-policy method. On-policy methods perform steadier updates. Off-policy updates show larger gradient norms in early steps, reflecting more abrupt shifts that influent generalization.

not isolate the independent contribution of weighting from that of negative trajectories. We hypothesize that advantage weighting concentrates updates on the tokens most responsible for task success, while learning from negatives suppresses reinforcement of incorrect patterns, and that together these encourage more useful exploration. Details are provided in Figure 13.

**KL regularization plays subtle roles**. On-policy RL performance remains largely unchanged with or without KL regularization. This suggests that, although KL regularization constrains models to stay closer to their original policy during updates, its overall effect on transferability is limited. One possible reason is that on-policy RL already aligns updates with the on-policy distribution, making the additional constraint from KL regularization less impactful.

# 6. Related Works

**Cross-domain improvement in Reasoning Models.** Interestingly, models heavily fine-tuned for formal reasoning sometimes falter on more general language tasks (Kumar et al., 2022). For example, OpenAI's o1, while excelling in STEM benchmarks, raised concerns about its versatility on other tasks (OpenAI, 2024). Follow-up research introduced reinforcement fine-tuning precisely to address this gap, aiming to adapt a generalist model's reasoning to new domains with limited data (Zhang et al., 2024). Indeed, o1 and similar reasoning models are built on strong general-purpose bases to retain broad knowledge (OpenAI, 2024; Hendrycks et al., 2021a). Nonetheless, trade-offs have been observed. Wang et al. (2024b) found that fine-tuning on a narrow set of instruction types can degrade a model's performance on other skills. Recent works have also stepped into analyzing the cross-domain performance of reasoning models (Sun et al., 2025), especially for RL-based approaches (Cheng et al., 2025; Hu et al., 2025).

**Representation-Level Analysis.** Fine-tuning for reasoning models not only boosts task performance but also alters the model's internal representations (Sheng et al., 2024). Recent studies have begun to probe how CoT-based fine-tuning changes the *latent space* of LLMs (Xu et al., 2025a; Wang

et al., 2025). Lobo et al. (2024) find that task-specific fine-tuning can reduce the faithfulness of a model's generated reasoning chains, indicating shifts in its underlying inference mechanisms. Complementary analyses of hidden states provide insight into such shifts. Xu et al. (2024) proposes a quantitative framework for assessing ideas that leverages hidden representations from LLMs to predict the merit of scientific ideas. Techniques like principal component analysis reveal that fine-tuning can carve out new directions in representation space that correspond to reasoning-related features (Xu et al., 2025a; Zhou et al., 2025).

# 7. Conclusion, Limitation and Discussion

We studied what drives *transferability* of reasoning models across reasoning and non-reasoning benchmarks. Our evidence shows that, beyond size and architecture, the fine-tuning paradigm is decisive: RL-tuned models improve math reasoning while preserving positive transfer to other reasoning and non-reasoning tasks, whereas SFT often induces negative transfer on non-reasoning benchmarks. Latent-space PCA further indicates that RL stays close to the backbone (stable features), while SFT yields larger shifts, especially on non-reasoning data. Token–distribution analysis aligns with this: RL selectively adjusts a small set of task-relevant tokens, whereas SFT perturbs many irrelevant ones. Notably, *UniReason-Qwen3-14B-RL* trained on 47K math examples achieves the strongest balance of reasoning gains and general-domain retention among compared models, supporting these conclusions. We then studied which factor in RL benefits generalization by a set of controlled study, and learned that sample distribution is the most important factor.

However, we acknowledge that our paper still has several limitations. For example, our control study could be extended beyond math to other domains such as coding or additional reasoning tasks, in order to provide a more comprehensive view of reasoning models' transferability. On the post-training side, we primarily analyze SFT vs. RL, on-policy vs. off-policy training, and the role of credit assignment and negative examples; future work could incorporate more integrated paradigms, for example SFT followed by RL; RL followed by SFT; or the four stage post-training used by Deepseek and Qwen3 on SFT-RL-SFT-RL, or multi-stage training pipelines to further validate our conclusions.

Going forward, we plan to extend this approach in three directions: (1) exploring hybrid recipes to further optimize the trade-off between reasoning and non-reasoning tasks, spanning our current math-based approach to more reasoning domains; (2) developing adaptive regularization strategies that account for backbone architecture and size to enhance transferability; and (3) proposing more comprehensive tools to evaluate transferability in multi-modal, long-context, and interactive generation settings.

## Impact Statement

This paper presents work whose goal is to advance the field of machine learning, specifically focusing on the transferability of reasoning capabilities in Large Language Models (LLMs). Our research highlights a critical trade-off in current post-training paradigms: while off-policy finetuning on reasoning models can drive metric improvement on specific benchmarks, it often comes at the cost of degrading general capabilities and instruction-following skills. In contrast, we demonstrate that on-policy learning methods can enhance reasoning while better preserving the model's general utility and latent space structure.

The broader impact of this work lies in guiding the community away from narrow benchmark overfitting and towards the development of more robust, versatile AI systems. While advancing reasoning capabilities carries potential dual-use risks common to general-purpose AI, our work primarily focuses on evaluation methodology and training stability to ensure these capabilities are developed reliably. We do not foresee specific negative societal consequences directly stemming from this study beyond the standard considerations of training large models.

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

# A. Appendix

## A.1. Ablations on Learning Rate, Regularization and Model Family

We conduct additional ablations to test whether our conclusions are sensitive to (i) the underlying model family and scale, (ii) the choice of lower learning rate for SFT, and (iii) the use of standard regularization such as L2-SP. We also expand evaluation to a broader set of additional benchmarks.

**Model family and learning rate.** Table 5 reports results on two additional base models, Qwen3-4B and Llama3.1-8B, under SFT and RL with both standard and reduced learning rates. Across both architectures, we observe the same qualitative pattern as in our main Qwen3-14B study: math-only SFT substantially improves math performance but often harms non-reasoning capabilities, whereas RL improves math and preserves (or slightly improves) non-reasoning tasks. These results indicate that our main conclusions are *not* specific to the Qwen3-14B family, but hold across distinct architectures and scales. A natural concern is that the larger representational drift under SFT might simply be caused by a higher learning rate. To probe this, we trained with a much smaller learning rate ($1 \times 10^{-6}$) on both Qwen3-4B and Llama3.1-8B. As shown in Table 5, reducing the learning rate uniformly degrades performance: for Qwen3-4B, *Math Avg.* drops from 42.6 to 34.7 and *Other Reasoning Avg.* from 31.3 to 22.3, while *Non-Reasoning Avg.* decreases slightly further from 25.3 to 22.9, remaining far below the base model. Similarly, for Llama3.1-8B, the low-learning-rate SFT model underperforms the standard SFT model on all three task families and still exhibits strong negative transfer on non-reasoning tasks. Thus, decreasing the SFT learning rate does not improve the performance of SFT; instead it primarily reduces math gains and weakens other reasoning performance. Combined with our on-policy vs. off-policy comparisons (which use the same SFT learning rate), this supports our claim that sampling distribution and credit assignment, rather than learning rate alone, are the main drivers of cross-domain transfer behavior.

**Regularization with L2-SP.** Table 6 evaluates Qwen3-8B under standard SFT and SFT with L2-SP regularization, in both off-policy and on-policy settings. The results show that L2-SP produces only minor changes in both raw averages and transferability indices. In the off-policy setting, non-reasoning performance is 26.6 for vanilla SFT and 26.4 for SFT+L2-SP, with $\text{TI}_{non} = -40.5$ vs. $-41.2$, indicating nearly the same degree of negative transfer. In the on-policy setting, SFT+L2-SP achieves slightly higher math and other reasoning averages than vanilla on-policy SFT, but *Non-Reasoning Avg.* remains very similar, and $\text{TI}_{non}$ changes only modestly. Consistent with our mechanistic analysis, what matters more for transfer is *where* and *how* updates are concentrated (on-policy sampling, advantage-like credit assignment, and negative examples), rather than merely changing their magnitude.

**Broader evaluation benchmarks.** Table 7 extends our evaluation to five additional benchmarks: PIQA (Bisk et al., 2020), CommonsenseQA (Talmor et al., 2019), RACE (Lai et al., 2017), and LogiQA (Liu et al., 2020), OpenBookQA (Mihaylov et al., 2018). These tasks are not used during math-only post-training and therefore test whether math specialization disrupts or preserves general capabilities. On all five benchmarks, the RL-tuned model UniReason-Qwen3-14B outperforms both the base model and the math-SFT variants. In contrast, the math-SFT models often hurt performance on non-reasoning benchmarks despite strong math improvements. These additional benchmarks corroborate our main finding: math-only RL can improve math reasoning without sacrificing, and in some cases improving, non-reasoning and commonsense skills, whereas standard math-SFT frequently induces negative transfer to such tasks.

**Broader Training Data.** To analyze the impact of broader training data, we additionally consider the General-Reasoner-14B model (Ma et al., 2025) as a case study. In contrast to our math-only setting (UniReason-Qwen3-14B), which is fine-tuned on $\sim 47\text{k}$ purely mathematical problems, General-Reasoner is trained on a WebInstruct-filtered corpus containing $\sim 230\text{k}$ diverse reasoning questions beyond math (e.g., physics, chemistry, business, etc.). To disentangle the effect of the training data from the optimization paradigm, we reproduce a SFT-finetuned model, **General-Reasoner-14B-SFT**, by training with our SFT pipeline on the authors' released WebInstruct-based dataset, and compare it to our model under the same evaluation protocol. As shown in Table 10, this result shows that our conclusion is consistent on not only math-only data. It can also be observed on different training data sources.

## A.2. Ablations on the Design of Transferability Index

We analyze the effect of our Transferability Index (TI) design and the signed square-root transformation. Recall that $\text{TI}_{other}$ and $\text{TI}_{non}$ quantify how much math gains transfer to other reasoning and non-reasoning task families, after normalizing for

*Table 5.* Performance of additional base models under SFT and RL. (1e-6) denotes the model is trained with a learning rate of $1 \times 10^{-6}$. Other models are trained with default settings mentioned in the paper.

| Model | Math Avg. | Other Reasoning Avg. | Non-Reasoning Avg. | $TI_{other}$ | $TI_{non}$ |
|---|---|---|---|---|---|
| Qwen3-4B-Base | 12.2 | 16.1 | 39.6 | | |
| Qwen3-4B-RL | 40.5 | 32.6 | 42.6 | 34.8 | 22.3 |
| Qwen3-4B-SFT | 42.6 | 31.3 | 25.3 | 32.7 | -40.6 |
| Qwen3-4B-SFT (1e-6) | 34.7 | 22.3 | 22.9 | 25.9 | -44.3 |
| Llama3.1-8B-Base | 3.8 | 8.9 | 24.7 | | |
| Llama3.1-8B-RL | 6.9 | 10.1 | 25.0 | 71.8 | 7.7 |
| Llama3.1-8B-SFT | 7.8 | 11.2 | 17.3 | 70.4 | -34.7 |
| Llama3.1-8B-SFT (1e-6) | 4.6 | 9.7 | 14.7 | 56.3 | -38.9 |

*Table 6.* Performance of Qwen3-8B variants with and without L2-SP regularization. $TI_{other}$ and $TI_{non}$ denote transferability indices for other reasoning and non-reasoning tasks.

| Model | Math Avg. | Other Reasoning Avg. | Non-Reasoning Avg. | $TI_{other}$ | $TI_{non}$ |
|---|---|---|---|---|---|
| Qwen3-8B-Base | 27.6 | 23.6 | 33.6 | | |
| Qwen3-8B-Off-policy-SFT | 41.9 | 34.4 | 26.6 | 18.3 | -40.5 |
| Qwen3-8B-Off-policy-SFT-L2SP | 42.8 | 35.1 | 26.4 | 16.9 | -41.2 |
| Qwen3-8B-On-policy-SFT | 33.7 | 35.7 | 35.0 | 68.6 | 30.2 |
| Qwen3-8B-On-policy-SFT-L2SP | 35.1 | 36.2 | 34.7 | 66.3 | 27.8 |

domain difficulty. We apply a signed square-root transform to compress very large positive/negative values while preserving the sign. Tables 8 and 9 report both the transformed values and the raw TI, together with their ranks. The key observation is that the transform does not change our qualitative conclusions: for $TI_{non}$, the ranking of models within RL and SFT groups is identical under both scales, and all SFT models still show negative transfer. For $TI_{other}$, the top RL models and the top SFT models remain the same, with only minor swaps among mid-ranked models that do not affect any claims in the paper. In all cases, the clear separation between RL models (mostly large positive TI) and many math-SFT models (often modest or negative TI) is preserved.

## A.3. Additional Related Work

**Supervised Fine-Tuning vs. Reinforcement Learning for LLMs.** Fine-tuning methods for reasoning typically fall into two major categories: supervised fine-tuning and reinforcement learning (Chen et al., 2024). SFT methods predominantly utilize annotated reasoning trajectories or solution traces, directly training models to replicate explicit reasoning sequences from datasets (Wei et al., 2022; Wang et al., 2023). RL-based fine-tuning, however, guides models by rewarding accurate and logically coherent reasoning steps without explicit step-by-step supervision, allowing exploration and optimization of reasoning pathways through feedback loops (Ziegler et al., 2019; Liu et al., 2025; Wang et al., 2024c; Chu et al., 2025).

**Reasoning Fine-Tuning of LLMs.** Recent advancements in large language models have notably emphasized specialized fine-tuning methods to enhance reasoning capabilities (Wong et al., 2025; Chen et al., 2023; Ziegler et al., 2019; Liu et al., 2025; Wang et al., 2024c; Li et al., 2025b; Feng et al., 2025; Xu et al., 2025b; Yang et al., 2025; Li et al., 2025a; Yeo et al., 2025). The chain-of-thought prompting strategy introduced by Wei et al. (2022) encourages models to produce step-by-step explanations, significantly boosting performance in symbolic reasoning tasks (Lambert et al., 2025; Wei et al., 2022; Longpre et al., 2023; Yu et al., 2024). Subsequent extensions, such as DeepSeek-R1 (Team, 2025a), have integrated reinforcement learning approaches alongside CoT, optimizing models through reward-driven policy improvements. Such RL-enhanced fine-tuning has achieved state-of-the-art results on benchmarks and competitive programming challenges (Hendrycks et al., 2021b; Team et al., 2025; Team, 2025a; Lambert et al., 2025).

*Table 7.* Performances of models in our control study on additional non-reasoning benchmarks.

| Model | PIQA | LogiQA | RACE | CommonsenseQA | OpenBookQA |
|---|---|---|---|---|---|
| Qwen3-14B Base | 79.5 | 32.7 | 40.2 | 50.9 | 32.0 |
| Qwen3-14B think | 78.5 | 28.4 | 38.0 | 20.6 | 25.6 |
| Qwen3-14B no-think | 74.3 | 26.7 | 39.1 | 19.2 | 28.4 |
| UniReason-Qwen3-14B (RL) | 82.4 | 35.8 | 43.3 | 55.2 | 35.7 |

*Table 8.* Transferability Index for other reasoning tasks. We report both the signed square-root transformed value and the raw $\text{TI}_{\text{other}}$, together with their ranks within RL or SFT models.

| Alias | Category | $\text{TI}_{\text{other}}$ (sqrt) | Rank (sqrt) | $\text{TI}_{\text{other}}$ | Rank |
|---|---|---|---|---|---|
| General-Reasoner-RL-14B | RL | 125.1 | 1 | 214.5 | 1 |
| SimpleRL-14B | RL | 98.6 | 2 | 188.0 | 2 |
| SimpleRL-1.5B | RL | 88.1 | 3 | 67.2 | 4 |
| UniReason-RL-14B | RL | 84.7 | 4 | 82.4 | 3 |
| DAPO-32B | RL | 62.3 | 5 | 55.2 | 5 |
| SimpleRL-Llama3.1-8B | RL | 57.0 | 6 | 23.9 | 6 |
| SimpleRL-7B | RL | 15.8 | 7 | 15.3 | 7 |
| Deepseek-R1-Distill-Qwen-7B | SFT | 62.1 | 1 | 59.0 | 2 |
| UniReason-SFT-14B-think | SFT | 53.7 | 2 | 75.4 | 1 |
| OpenThinker3-7B | SFT | 45.7 | 3 | 49.3 | 3 |
| LIMO-32B | SFT | 41.6 | 4 | 33.0 | 4 |
| OpenThinker2-32B | SFT | 40.2 | 5 | 31.8 | 5 |
| S1.1-32B | SFT | 34.7 | 6 | 26.7 | 6 |
| Mistral-Reason-24B | SFT | 23.3 | 7 | 11.7 | 7 |
| S1.1-7B | SFT | -67.3 | 8 | -87.4 | 8 |

## A.4. Detailed Evidence of Our Observed Phenomena

As discussed in Section 2.1, we provided the complete evaluation for the Transferability Index for the off-the-shelf models on other reasoning and non-reasoning tasks in Table 11.

## A.5. Full Evaluation Setup

### A.5.1. POST-TRAINING METHODS

**Reinforcement Learning** has recently proven effective at steering large language models toward complex, multi-step objectives by optimizing policies with scalar reward signals (Zeng et al., 2025). We used the Verl framework(Sheng et al., 2025) and GRPO (Shao et al., 2024) setup to fine-tune the Qwen-3-14B-Base model, utilizing answer correctness as the reward signal. Our RL uses a learning rate of $1 \times 10^{-6}$ with an overall train batch size of 512 and clipping thresholds set between 0.22 and 0.28. We generate sequences up to 16k tokens long and perform 16 rollouts per prompt, then update the model in mini-batches of 128 samples. Both KL-divergence and entropy penalties are turned off (coefficients set to zero). We train the model for 140 steps and used the corresponding checkpoint.

**Supervised Fine-Tuning** remains a fundamental technique for adapting large pre-trained models by directly minimizing cross-entropy on high-quality datasets (Chu et al., 2025). We use the LLaMA-Factory framework (Zheng et al., 2024), which is an extensible and user-friendly framework supporting multiple architectures and advanced optimization algorithms, to fine-tune our model on teacher-generated chain-of-thought traces. We use $5 \times 10^{-5}$ as learning rate, the batch size is 512 and we train for 1.5 epoch to align with our RL settings.

*Table 9.* Transferability Index for non-reasoning tasks. We report both the signed square-root transformed value and the raw $TI_{non}$, together with their ranks within RL or SFT models.

| Alias | Category | $TI_{non}$ (sqrt) | Rank (sqrt) | $TI_{non}$ | Rank |
|---|---|---|---|---|---|
| SimpleRL-14B | RL | 103.8 | 1 | 183.5 | 1 |
| General-Reasoner-RL-14B | RL | 68.0 | 2 | 59.1 | 2 |
| UniReason-RL-14B | RL | 53.4 | 3 | 30.5 | 3 |
| SimpleRL-7B | RL | 35.2 | 4 | 28.5 | 4 |
| SimpleRL-1.5B | RL | 9.8 | 5 | 1.5 | 6 |
| SimpleRL-Llama3.1-8B | RL | 5.4 | 6 | 3.3 | 5 |
| DAPO-32B | RL | -40.4 | 7 | -20.4 | 7 |
| Deepseek-R1-Distill-Qwen-7B | SFT | -4.2 | 1 | -0.4 | 1 |
| LIMO-32B | SFT | -27.9 | 2 | -7.9 | 2 |
| OpenThinker2-32B | SFT | -54.7 | 3 | -32.6 | 4 |
| S1.1-32B | SFT | -63.6 | 4 | -41.2 | 5 |
| UniReason-SFT-14B-think | SFT | -64.7 | 5 | -31.8 | 3 |
| OpenThinker3-7B | SFT | -71.6 | 6 | -59.2 | 6 |
| Mistral-Reason-24B | SFT | -91.5 | 7 | -86.2 | 7 |
| S1.1-7B | SFT | -218.2 | 8 | -326.8 | 8 |

*Table 10.* Additional Study on broader training data show that our finding can also be used outside of math-only data. A.1.

| **Math Reasoning** | | | | | | |
|---|---|---|---|---|---|---|
| **Model** | **AIME24** | **AIME25** | **MATH500** | **Olympiad** | **Average** | |
| Qwen3-14B-Base | 13.0 | 9.3 | 60.4 | 27.9 | 27.7 | - |
| General-Reasoner-14B-SFT | 35.0 | 22.6 | 78.4 | 22.5 | 39.6 | - |
| General-Reasoner-14B-RL | 24.4 | 19.2 | 83.0 | 32.3 | 39.7 | - |
| $\Delta$ (RL - SFT) | -10.6 | -3.4 | +4.6 | +9.8 | +0.1 | - |
| **Other Reasoning** | | | | | | |
| **Model** | **GPQA** | **LiveCodeBench2** | **ACPBench** | **HeadQA** | **Average** | $TI_{other}$ |
| Qwen3-14B-Base | 42.6 | 29.7 | 10.7 | 37.6 | 30.2 | - |
| General-Reasoner-14B-SFT | 43.9 | 19.7 | 64.3 | 39.9 | 41.1 | +52.2 |
| General-Reasoner-14B-RL | 56.1 | 32.9 | 75.0 | 40.0 | 51.0 | +121.6 |
| $\Delta$ (RL - SFT) | +12.2 | +13.2 | +10.7 | +0.1 | +9.1 | - |
| **Non-Reasoning** | | | | | | |
| **Model** | **CoQA** | **MC-TACO** | **IFEval** | **HaluEval** | **Average** | $TI_{non}$ |
| Qwen3-14B-Base | 10.0 | 67.7 | 69.2 | 35.7 | 45.7 | - |
| General-Reasoner-14B-SFT | 8.7 | 38.2 | 31.7 | 10.8 | 22.3 | -110.7 |
| General-Reasoner-14B-RL | 11.2 | 74.0 | 72.0 | 55.7 | 53.2 | +66.5 |
| $\Delta$ (RL - SFT) | +2.5 | +35.8 | +40.3 | +44.9 | +30.9 | - |

### A.5.2. TRAINING DATASETS

As briefed in Section 2.2, our base training dataset is a curated set of 47K high-quality mathematics problems. We stratified the examples using two complementary sources: low-difficulty problems drawn from the DeepScaler dataset (Luo et al., 2025), and high-difficulty (levels 3–5) problems extracted from SimpleRL (Zeng et al., 2025). To generate CoT annotations, we prompt each problem into the Qwen3-32B-Instruct model (Team, 2025b) and use reject sampling to generate our dataset.

To further explore the effect of training data distribution for SFT-based reasoning models, we also distill a larger and more comprehensive dataset collected from General-Reasoner (Ma et al., 2025), which contains 232K examples across reasoning and non-reasoning tasks (e.g., Math, Chemistry, Business). This additional distilled set is used to train the General-Reasoner model using supervised fine-tuning.

### A.5.3. BASELINES

In our experiments, we compare against Qwen3-14B-Base model (Team, 2025b), which is the original Qwen3-14B model without any further adaptation. This serves as the unmodified backbone for all fine-tuning models. Also, we report the results of Qwen3-14B-Instruct model under our tested benchmarks. It is an instruction-tuned version of Qwen3-14B model trained on a large, general-purpose instruction-following dataset. We evaluate it under two prompting modes:

- *think*: prompts include a special `<think>` token to explicitly elicit chain-of-thought reasoning.

- *no-think*: prompts are provided without the `<think>` token, relying solely on direct instruction-following behavior.

Due to the wide range and enormous training data, this model is considered to give the optimal outputs across tasks in the current 14B-series models.

To further validate our observation, we apply our controlled study pipeline also for General-Reasoner (Ma et al., 2025), a reasoning model tuned by cold RL with training data containing not only math but more diverse domains. We distilled the dataset for SFT finetuning using the same rejection sampling method with their proposed dataset. The dataset contains 232K samples covering various reasoning and non-reasoning tasks. Then we finetune the Qwen3-14B base model using the distilled dataset and name the model as General-Reasoner-Qwen3-14B(SFT) to directly compare with the RL-based General-Reasoner for a fairer and more comprehensive controlled study towards SFT and RL. The Transferability Index results could be retrieved in Table 11, the observation also confirms our initial hypothesis.

### A.5.4. EVALUATION BENCHMARKS

In the experiment, we evaluated our model across a wide range of benchmarks. Notably, to explicitly reveal the transferability of reasoning models, we grouped them into three categories by their content:

**Math Reasoning Datasets**   We collected the following datasets that are composed of mathematical problems, which means that they typically need a mathematical reasoning process to get the answer:

- **MATH500** (Hendrycks et al., 2021b): A curated subset of 500 problems sampled from the broader MATH dataset, covering topics like algebra, combinatorics, geometry, and number theory.

- **AIME**: Problems drawn from the American Invitational Mathematics Examination (AIME) 2024 and 2025, each with 30 challenging short-answer questions requiring multi-step reasoning.

- **OlympiadBench** (He et al., 2024): Problems sourced from international olympiads (e.g., IMO and regional contests).

**Other Reasoning Datasets**   We collected the following datasets that are mainly composed of general reasoning problems containing a wider range of subjects:

- **LiveCodeBench** (Jain et al., 2025): It is a continuously updated, contamination-free coding benchmark. We used its second version.

- **GPQA-Diamond**(Rein et al., 2024): It is a graduate-level question-answering dataset that contains multiple-choice questions in biology, physics, and chemistry. We followed its diamond split.

- **ACPBench** (Kokel et al., 2025): It has 7 atomic reasoning tasks around 13 classical planning domains. We only used the multiple-choice problems.

- **HeadQA** (Vilares & Gómez-Rodríguez, 2019): Multiple-choice QA from healthcare-specialist certification exams, including questions across pharmacology, chemistry, nursery, psychology, biology, and medicine.

**Non-reasoning Datasets**   We collected the following datasets that are mainly composed of problems with factual answers, which means that they do not need a reasoning process to give the answer:

*Table 11.* Comparison of model transferability indices on other reasoning and non-reasoning tasks across different training paradigms (SFT vs. RL), model sizes (7B vs. 32B), and model families (Qwen vs. Llama).

| Model | Base Model | Size | Method | $TI_{other}$ | $TI_{non}$ |
|---|---|---|---|---|---|
| **SFT vs RL** | | | | | |
| Deepseek-R1-Distill-Qwen-7B | Qwen2.5-Math-7B-Base | 7B | SFT | 60.4 | -4.2 |
| OpenThinker2-7B | Qwen2.5-7B-Instruct | 7B | SFT | 52.0 | -76.2 |
| OpenThinker3-7B | Qwen2.5-7B-Instruct | 7B | SFT | 44.5 | -70.0 |
| S1.1-7B | Qwen2.5-7B-Instruct | 7B | SFT | -65.5 | -213.3 |
| General-Reasoner-Qwen3-14B (SFT) | Qwen3-14B-Base | 14B | SFT | 77.8 | -127.6 |
| Mistral-Small-24B-Instruct-2501-Reasoning | Mistral-Small-24B-Instruct-2501 | 24B | SFT | 22.6 | -89.5 |
| OpenThinker2-32B | Qwen2.5-32B-Instruct | 32B | SFT | 39.1 | -53.5 |
| S1.1-32B | Qwen2.5-32B-Instruct | 32B | SFT | 33.8 | -62.2 |
| LIMO-32B | Qwen2.5-32B-Instruct | 32B | SFT | 40.5 | -27.3 |
| Qwen2.5-1.5B-SimpleRL | Qwen2.5-1.5B-Base | 1.5B | RL | 85.6 | 9.5 |
| SimpleRL-7B | Qwen2.5-7B-Base | 7B | RL | 15.4 | 34.4 |
| Qwen2.5-14B-SimpleRL | Qwen2.5-14B-Base | 14B | RL | 95.9 | 101.5 |
| General-Reasoner-Qwen3-14B (RL) | Qwen3-14B-Base | 14B | RL | 121.6 | 66.5 |
| DAPO-Qwen-32B | Qwen2.5-32B | 32B | RL | 60.7 | -39.5 |
| **7B vs 32B** | | | | | |
| S1.1-7B | Qwen2.5-7B-Instruct | 7B | SFT | -65.5 | -213.3 |
| S1.1-32B | Qwen2.5-32B-Instruct | 32B | SFT | 33.8 | -62.2 |
| OpenThinker2-7B | Qwen2.5-7B-Instruct | 7B | SFT | 52.0 | -76.2 |
| OpenThinker2-32B | Qwen2.5-32B-Instruct | 32B | SFT | 39.1 | -53.5 |
| **Qwen vs Llama** | | | | | |
| Qwen2.5-7B-SimpleRL | Qwen2.5-7B-Base | 7B | RL | 15.4 | 34.4 |
| Llama3.1-8B-SimpleRL | Llama3.1-8B | 8B | RL | 72.6 | 7.0 |

- **CoQA** (Reddy et al., 2019): It has 127K questions in dialogues over passages, focusing on maintaining context and coreference across turns.

- **IFEval** (Zhou et al., 2023): It contains over 500 prompts, each embedding verifiable instructions. Evaluates strict vs. loose adherence to instructions.

- **HaluEval** (Li et al., 2023): It contains human-annotated samples where models must distinguish factual content from hallucinations.

- **MC-TACO** (Zhou et al., 2019): It is a multiple-choice benchmark designed to evaluate models' temporal commonsense, covering duration, ordering, typical time, frequency, and stationarity.

### A.5.5. EVALUATION METRICS

We used LLM-Harness (Gao et al., 2024) to evaluate the models' performance on OlympiadBench, ACPBench, HeadQA, CoQA, HaluEval, MC-TACO and used Eval-Chemy (Raoof et al., 2025) MATH500, AIME24, AIME25, GPQA-Diamond, LiveCodeBench, IFEval. On MATH500, AIME24, AIME25, GPQA-Diamond, and LiveCodeBench, we used 0.6 as temperature, and 0.95 as top-p value. In our experiments, we used **accuracy** to evaluate the models' performance. Specifically, for AIME24 and AIME 25, we averaged accuracy on 10 samples. For GPQA-Diamond, LiveCodeBench and MATH 500, our score is the average accuracy over 3 samples. Specifically, we used version 2 and overall accuracy on LiveCodeBench. For ACPBench, we only used multiple choices, and averaged the score for all 10 tasks as the final score. For OlympiadBench, we only used math queries in English, and thus categorized Olympiad as a math benchmark. For HaluEval, the performance is the accuracy averaged on 3 tasks with zero-shot. And for IFEval, we used strict instruction accuracy as the score. For OlympiadBench, ACPBench, HeadQA, CoQA, HaluEval, and IFEval, we used greedy sampling and sampled only once.

*Table 12.* Comparison of mean PCA shift magnitudes on math, reasoning, and non-reasoning tasks across different training paradigms (SFT vs. RL), model sizes (7B vs. 32B), and model families (Qwen vs. Llama). **Red-highlighted rows** indicate models trained with RL, non-Qwen baselines, or models with larger size.

| Model | Base Model | Size | Method | Math | Other | Non |
|---|---|---|---|---|---|---|
| **SFT vs RL** | | | | | | |
| Deepseek-R1-Distill-Qwen-7B | Qwen2.5-Math-7B | 7B | SFT | 38.8 | 27.4 | 42.2 |
| OpenThinker2-7B | Qwen2.5-7B-Instruct | 7B | SFT | 388.8 | 444.0 | 5486.2 |
| OpenThinker3-7B | Qwen2.5-7B-Instruct | 7B | SFT | 409.9 | 484.2 | 5228.2 |
| S1.1-7B | Qwen2.5-7B-Instruct | 7B | SFT | 9.8 | 15.5 | 205.7 |
| Qwen3-14B | Qwen3-14B-Base | 14B | SFT | 40.4 | 13.9 | 129.8 |
| General-Reasoner-Qwen3-14B (SFT) | Qwen3-14B-Base | 14B | SFT | 7.9 | 10.5 | 118.2 |
| OpenThinker2-32B | Qwen2.5-32B-Instruct | 32B | SFT | 25.4 | 72.0 | 726.8 |
| S1.1-32B | Qwen2.5-32B-Instruct | 32B | SFT | 2.4 | 1.8 | 7.5 |
| LIMO-32B | Qwen2.5-32B-Instruct | 32B | SFT | 15.7 | 9.0 | 7.5 |
| Qwen2.5-1.5B-SimpleRL-Zoo | Qwen2.5-1.5B | 1.5B | RL | 0.5 | 0.3 | 0.6 |
| Qwen2.5-7B-SimpleRL-Zoo | Qwen2.5-7B | 7B | RL | 0.6 | 0.2 | 0.6 |
| Llama-3.1-8B-SimpleRL-Zoo | Llama-3.1-8B | 8B | RL | 0.3 | 0.1 | 0.0 |
| Qwen2.5-14B-SimpleRL-Zoo | Qwen2.5-14B | 14B | RL | 0.9 | 1.0 | 2.7 |
| General-Reasoner-Qwen3-14B(RL) | Qwen3-14B-Base | 14B | RL | 2.9 | 4.6 | 64.8 |
| DAPO-Qwen-32B | Qwen2.5-32B | 32B | RL | 2.6 | 1.1 | 1.1 |
| **7B vs 32B** | | | | | | |
| S1.1-7B | Qwen2.5-7B-Instruct | 7B | SFT | 9.8 | 15.5 | 205.7 |
| S1.1-32B | Qwen2.5-32B-Instruct | 32B | SFT | 2.4 | 1.8 | 7.5 |
| OpenThinker2-7B | Qwen2.5-7B-Instruct | 7B | SFT | 388.8 | 444.0 | 5486.2 |
| OpenThinker2-32B | Qwen2.5-32B-Instruct | 32B | SFT | 25.4 | 72.0 | 726.8 |
| **Qwen vs Llama** | | | | | | |
| Qwen2.5-7B-SimpleRL-Zoo | Qwen2.5-7B | 7B | RL | 0.6 | 0.2 | 0.6 |
| Qwen3-14B-Base-SimpleRL-Zoo | Qwen3-14B-Base | 14B | RL | 0.9 | 1.0 | 2.7 |
| Llama-3.1-8B-SimpleRL-Zoo | Llama-3.1-8B | 8B | RL | 0.3 | 0.1 | 0.0 |

**A.6. PCA Analysis under Varying Settings**

Table 12 summarizes the mean $d^{(*)}$ across math, other-reasoning, and non-reasoning tasks, providing an overall assessment of latent-space shifts under different training paradigms. Figure 6, 7, 8, 9 illustrates the paradigm comparison between SFT and RL, Figure 10 contrasts model sizes (7B versus 32B), and Figure 11 compares model families (Qwen vs. Llama).

**Impact of Training Paradigm.** RL-based fine-tuning consistently results in lower PCA shifts than SFT across math, other-reasoning, and non-reasoning tasks. As shown in Table 12, models such as SimpleRL-7B and SimpleRL-14B exhibit significantly smaller feature shifts compared to their SFT-trained counterparts. This observation is further visualized in Figure 6, 7, 8, 9, which demonstrates more concentrated representation shifts under RL. These findings are consistent with the phenomena discussed in Section 2.1, reinforcing that RL enhances generalization by better preserving internal representations. Overall, these results suggest that RL is substantially more effective at preserving general-domain representations and mitigating catastrophic forgetting.

**Effect of Model Size.** Increasing model size from 7B to 32B substantially reduces PCA shifts on math, other-reasoning, and non-reasoning tasks, as shown in Table 12. Figure 10 further confirms that larger models exhibit more concentrated and stable latent representations in non-reasoning domains, indicating reduced feature drift. These findings suggest that scaling up model size is particularly effective for enhancing stability and resistance to latent space drift in general-domain tasks.

**Effect of Model Family.** The results in Table 12 and Figure 11 indicate that both RL-tuned Llama and Qwen models exhibit small PCA shifts. While Qwen displays similar shifts as Llama on reasoning tasks, it also has higher drift on other-reasoning and non-reasoning tasks. In contrast, Llama models show greater resilience to drift in general domains. These differences stem from the distinct architectures and pre-training data, suggesting that RL strategies should be tailored to each model's characteristics.

**Summary.** Our analyses show that latent space drift is mainly determined by training paradigm, model size, and model family. RL-based fine-tuning is the most effective at preserving general-domain representations and mitigating catastrophic forgetting. While larger models offer greater feature stability, especially on other-reasoning and non-reasoning tasks, scaling alone is insufficient. Qwen models remain sensitive to reasoning-related pre-training data, underscoring the need for tailored strategies. Overall, RL-based training is critical for developing robust language models that balance general-domain coverage with reasoning ability.

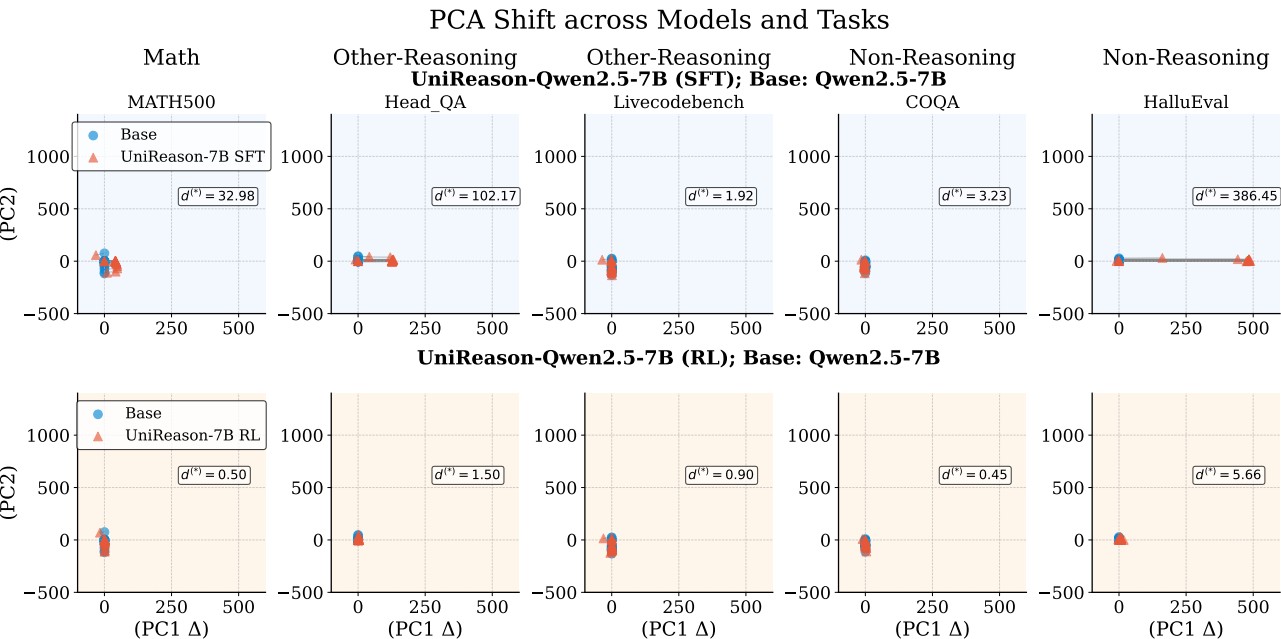

*Figure 6.* PCA shift of Qwen2.5-7B across different training methods and tasks. $d^{(*)}$ is the Euclidean distance between representation centroids before and after training. The first row shows models trained with SFT, and the last row shows models trained with RL. RL training results in the smallest PCA shift for all task types, suggesting more stable latent representations.

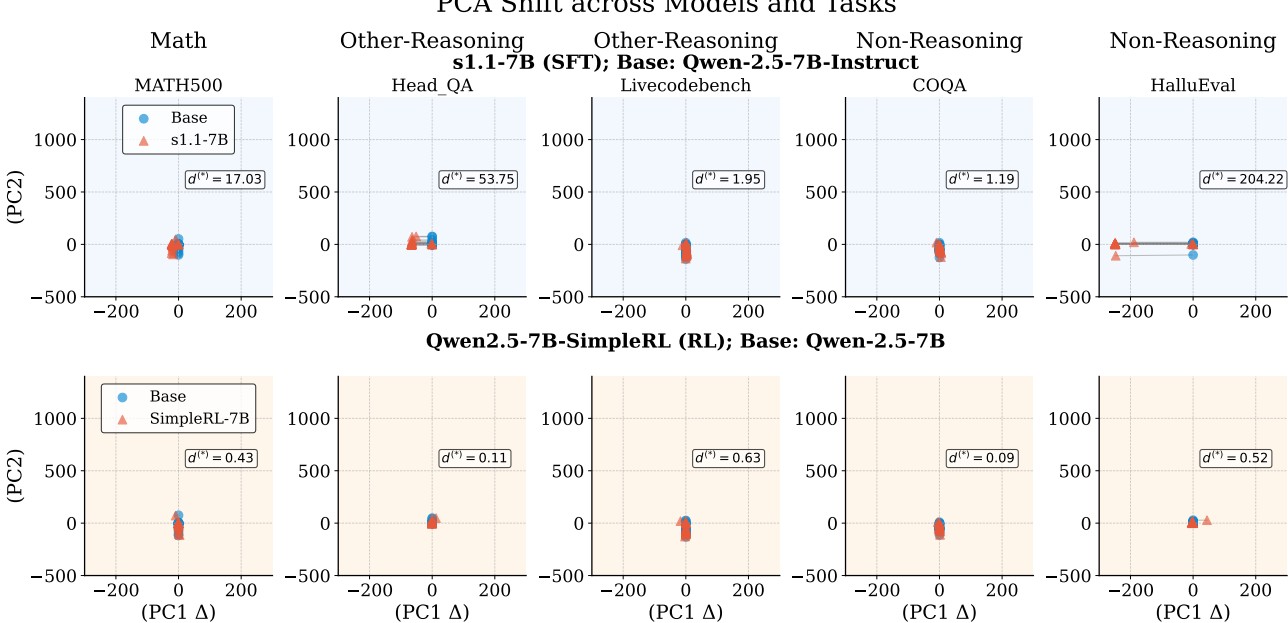

*Figure 7.* PCA shift for Qwen-2.5-7B-Instruct and Qwen2.5-7B under different training paradigms and benchmarks. The first row corresponds to models trained with SFT, while the second row corresponds to the RL. The RL-based training paradigm achieves the lowest PCA shift across all task categories, indicating enhanced stability in the model's latent representation space for math, other reasoning, and non-reasoning tasks. $d^{(*)}$ denotes the Euclidean distance between the centroids of representations before and after training.

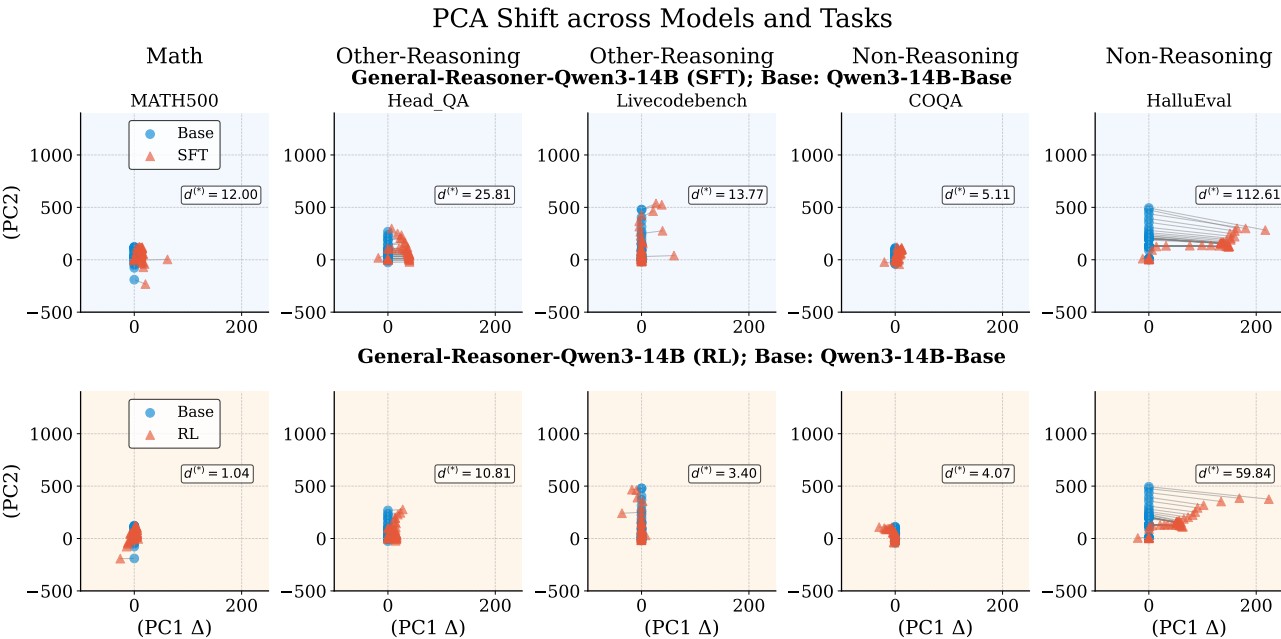

*Figure 8.* PCA shift for Qwen3-14B-Base under different training paradigms and benchmarks. The first row corresponds to models trained with SFT, while the second row corresponds to the RL. The RL-based training paradigm achieves the lowest PCA shift across all task categories, indicating enhanced stability in the model's latent representation space for math, other reasoning, and non-reasoning tasks. $d^{(*)}$ denotes the Euclidean distance between the centroids of representations before and after training.

## A.7. Details on Ablation Study

### A.7.1. GRADIENT NORM OF SFT AND RL

As shown in Figure 12 and Table 13, RL and SFT exhibit distinct gradient dynamics. RL performs gentler early updates (Early Slope closer to zero) without the sharp spikes observed in SFT, and accumulates smaller overall movement in

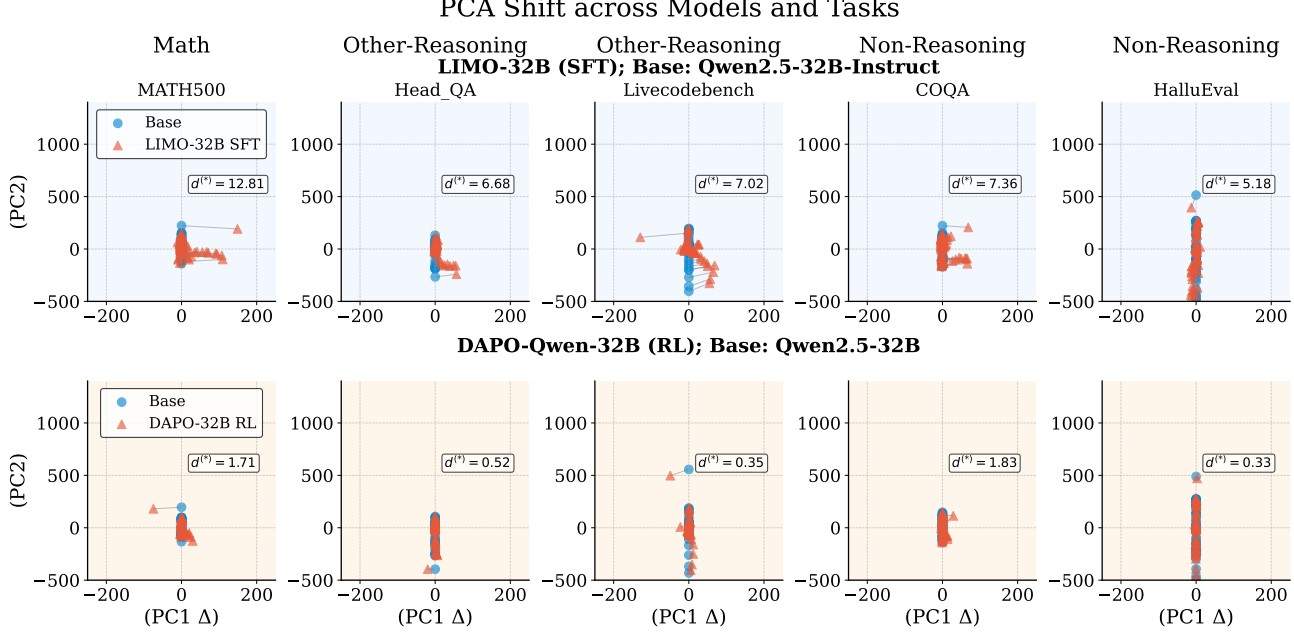

*Figure 9.* PCA shift for Qwen2.5-32B-Instruct and Qwen2.5-32B under different training paradigms and benchmarks. The first row corresponds to models trained with SFT, while the second row corresponds to the RL. The RL-based training paradigm achieves the lowest PCA shift across all task categories, indicating enhanced stability in the model's latent representation space for math, other reasoning, and non-reasoning tasks. $d^{(*)}$ denotes the Euclidean distance between the centroids of representations before and after training.

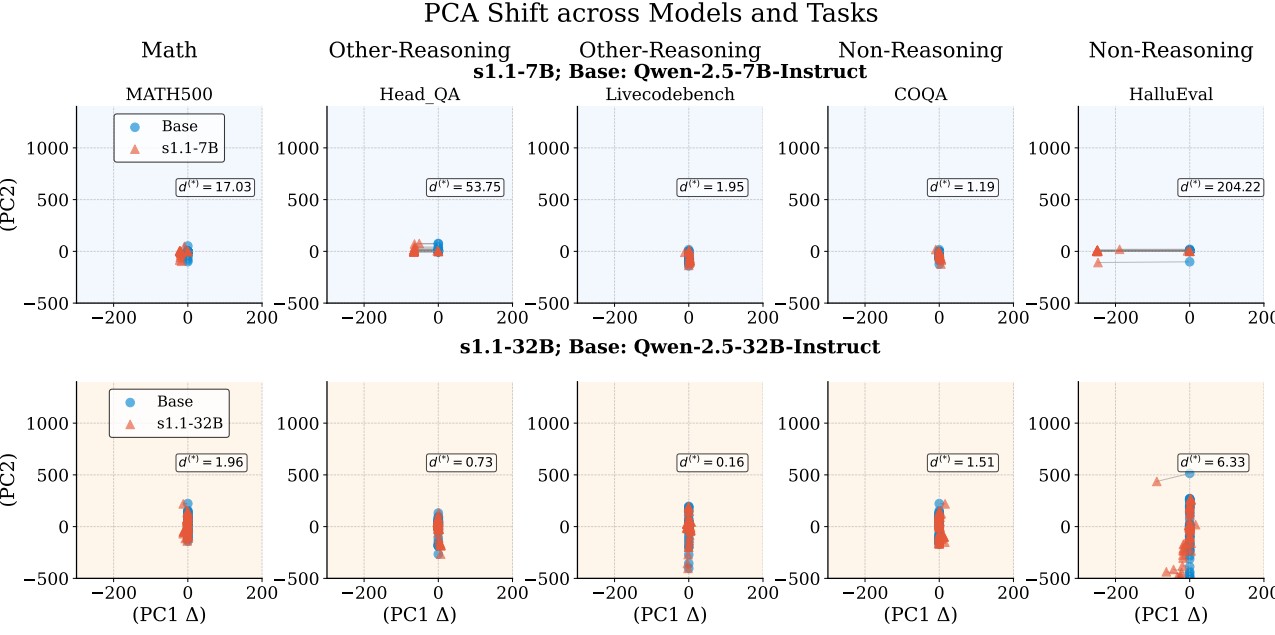

*Figure 10.* PCA shift for under different model sizes (7B vs 32B) and benchmarks. The first row corresponds to the 7B model, while the second row corresponds to the 32B model. $d^{(*)}$ denotes the Euclidean distance between the centroids of representations before and after training.

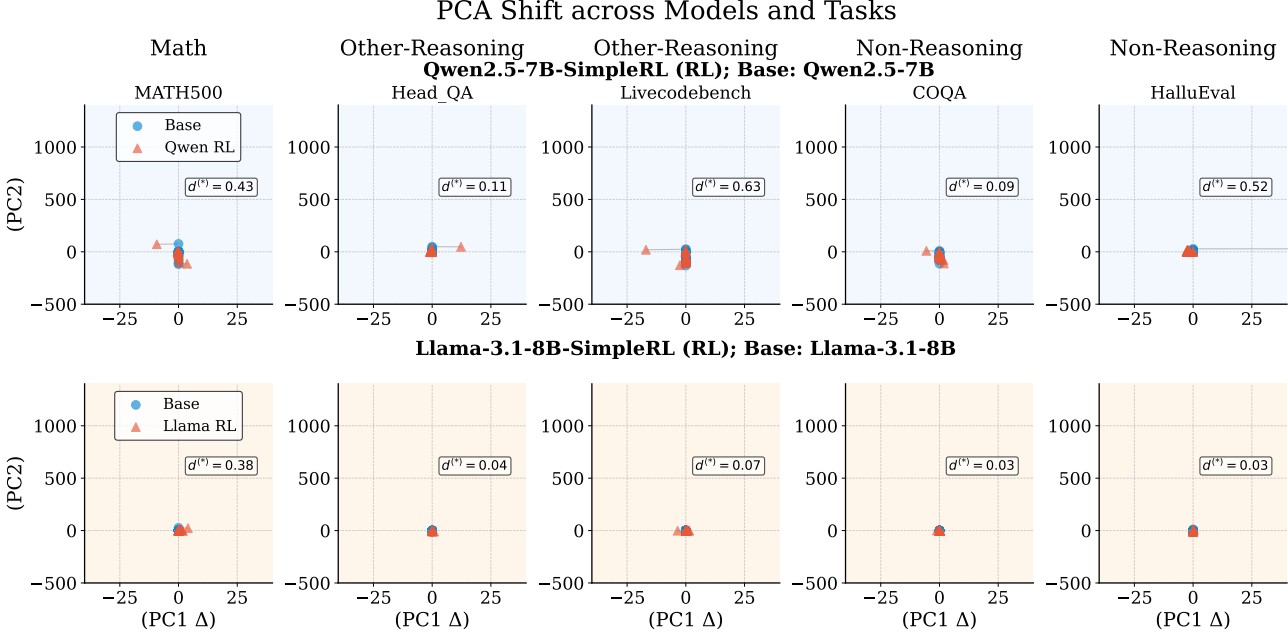

*Figure 11.* PCA shift under different model families (Qwen vs Llama) and benchmarks. The first row corresponds to the Qwen model, while the second row corresponds to the Llama model. $d^{(*)}$ denotes the Euclidean distance between the centroids of representations before and after training.

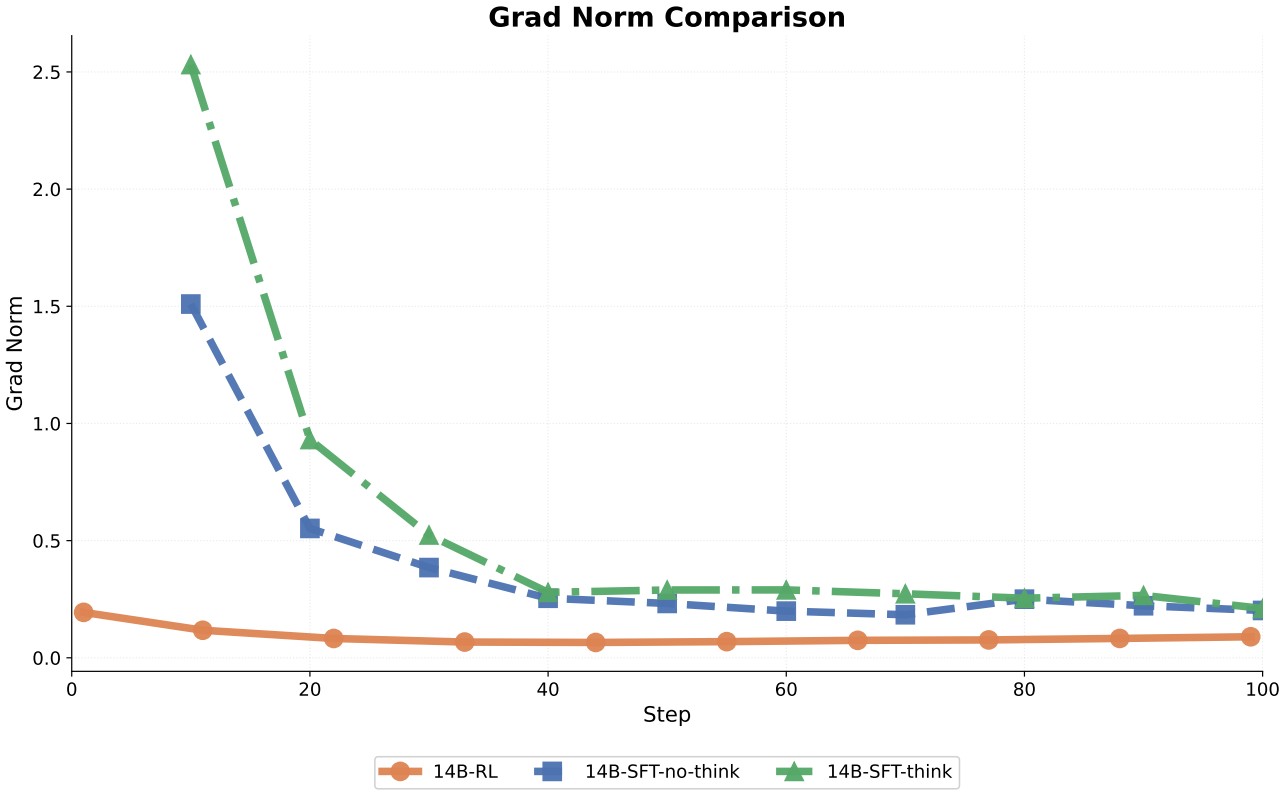

*Figure 12.* Grad norm comparison across different fine-tuning strategies for the first 100 steps. RL exhibits consistently lower gradient norms compared to both SFT-no-think and SFT-think models, suggesting more stable optimization dynamics during fine-tuning.

| Model | Type | Early Slope | $t_{1/2}$ | Total Grad. | Late Avg. | Late Var. | Decay Rate | Oscillation |
|---|---|---|---|---|---|---|---|---|
| 14B-RL | RL | $-0.005$ | 22.0 | 48.190 | 0.448 | 0.041 | $-0.262$ | 0.034 |
| 14B-SFT-no-think | SFT | $-0.019$ | 80.0 | 64.906 | 0.405 | 0.098 | $-0.592$ | 0.368 |
| 14B-SFT-think | SFT | $-0.030$ | 80.0 | 64.187 | 0.383 | 0.112 | $-0.625$ | 0.382 |

*Table 13.* **Gradient norm indicators in early training (first 100 steps).** Smaller *Early Slope* values (closer to zero) show more gradual early updates. Lower *Total Grad.* suggests smaller overall movement in parameter space. Higher *Late Avg.* with lower *Late Var.* reflects a stable learning signal in later steps. A less negative *Decay Rate* shows slower reduction of gradient strength. Finally, smaller *Oscillation* values mean smoother training, with fewer rapid shifts. Across all metrics, RL shows more stable and generalizable training behavior than SFT.

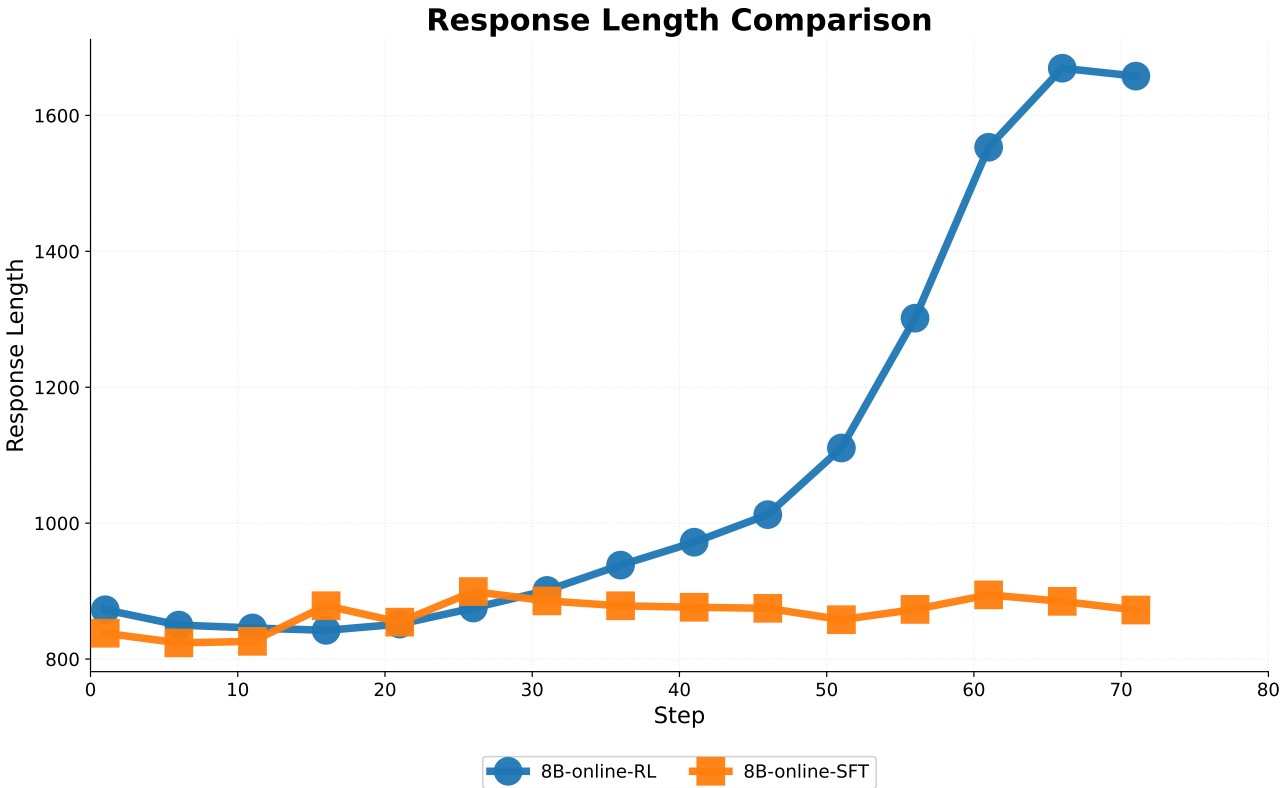

*Figure 13.* Response length comparison across online RL and SFT fine-tuning (first 80 steps). Online RL gradually increases response length during training, while SFT remains stable. This indicates that RL promotes stronger exploration, supported by mechanisms such as negative examples and credit assignment that guide the model to generate more diverse and extended outputs. Longer responses reflect broader policy exploration, which is often linked to better generalization and transfer across tasks.

parameter space (lower Total Grad.). In later steps, RL maintains a stronger and more stable gradient signal (higher Late Avg., lower Late Var.), with a slower decay rate and much less oscillation. In contrast, SFT shows large initial fluctuations followed by rapid collapse toward near-zero gradients. Overall, RL demonstrates a more stable and persistent optimization process, whereas SFT tends to quickly saturate and lose effective gradient signals.

### A.7.2. RESPONSE LENGTH OF ONLINE RL AND ONLINE SFT

Compared to SFT, online RL produces a marked increase in response length as training progresses. This pattern reflects RL's stronger exploration ability: negative examples discourage uninformative outputs, while credit assignment amplifies signals from task-relevant tokens. Together, these mechanisms encourage the model to explore a wider solution space, leading to longer and more varied responses. Such exploration is important for improving transferability, as it enables the model to adapt its reasoning strategies beyond the narrow supervision provided in SFT.

## A.8. Token Distribution Shifts

### A.8.1. CASE STUDY AND DETAILS

This part provides the complete queries and responses in our case study in Section 4. Table 14 enumerates all shifted tokens for both reasoning and non-reasoning queries. RL modifies a compact set of task-salient tokens, whereas SFT shifts a much larger set including many off-topic tokens.

Case Study:

**Reasoning Task**

**Query:**
Solve the following math problem. Present the final answer in the format: Final Answer: boxed{your_answer}. Problem: Ten treeks weigh as much as three squigs and one goolee. Two treeks and one goolee are equal in weight to one squig. The combined weight of how many treeks equals the weight of one squig?

**RL-tuned Models:**
Let's define the variables for the weights of the treeks, squigs, and goolees. Let: - $T$ be the weight of one treek, - $S$ be the weight of one squig, - $G$ be the weight of one goolee.
We are given two pieces of information: 1. Ten treeks weigh as much as three squigs and one goolee: $10T = 3S + G$ 2. Two treeks and one goolee are equal in weight to one squig: $2T + G = S$
We need to find how many treeks combined equal the weight of one squig. In other words, we need to find a number $x$ such that $xT = S$.
First, let's solve the second equation for $G$: $2T + G = S \Rightarrow G = S - 2T$
Now, substitute $G = S - 2T$ into the first equation: $10T = 3S + (S - 2T)$ $10T = 3S + S - 2T$ $10T = 4S - 2T$
Now, add $2T$ to both sides to isolate the terms involving $S$: $10T + 2T = 4S$ $12T = 4S$
Now, divide both sides by 4 to solve for $S$: $3T = S$
This tells us that the weight of one squig is equal to the weight of three treeks. Therefore, the combined weight of three treeks equals the weight of one squig.
Final Answer: $\boxed{3}$

**SFT-tuned Models:**

Okay, let's see. I need to solve this problem where they talk about treeks, squigs, and goolees. The question is asking how many treeks combined equal the weight of one squig.

First, let me parse the problem again. The first statement says ten treeks weigh as much as three squigs and one goolee. So, $10T = 3S + G$, where $T$ is treek, $S$ is squig, $G$ is goolee. The second statement is two treeks and one goolee equal one squig. So, $2T + G = S$. The question is asking for how many treeks equal one squig. So, find $x$ such that $xT = S$.

Alright, so I have two equations:

1) $10T = 3S + G$

2) $2T + G = S$

I need to solve these equations to find $x$ where $xT = S$. Let me see. Maybe I can express $G$ from the second equation and substitute into the first.

From equation 2: $G = S - 2T$. Let me check that. If $2T + G = S$, then subtract $2T$ from both sides: $G = S - 2T$. Yes, that's right.

Now substitute this into equation 1. So equation 1 becomes:

$10T = 3S + (S - 2T)$

Simplify the right side: $3S + S - 2T = 4S - 2T$

So now the equation is $10T = 4S - 2T$

Let me add $2T$ to both sides: $10T + 2T = 4S \Rightarrow 12T = 4S$

Divide both sides by 4: $3T = S$. Wait, so $S = 3T$?

Wait, hold on. The question is asking for how many treeks equal one squig. If $S = 3T$, then 1 squig equals 3 treeks. Where did I go wrong?

Let me check my equations again. The problem says:

First statement: Ten treeks weigh as much as three squigs and one goolee. So $10T = 3S + G$. Correct.

Second statement: Two treeks and one goolee are equal in weight to one squig. So $2T + G = S$. Correct.

Then solving for $G$ from the second equation: $G = S - 2T$. Substitute into the first equation:

$10T = 3S + (S - 2T) \Rightarrow 10T = 4S - 2T \Rightarrow 12T = 4S \Rightarrow 3T = S$. So $S = 3T$. Therefore, 1 squig equals 3 treeks. Maybe I misread the problem?

Wait, maybe the problem is asking for how many treeks equal one squig, but in the equations, maybe I have to express $T$ in terms of $S$. Let me try solving for $T$. From $S = 3T$, then $T = \frac{S}{3}$. So one treek is $\frac{1}{3}$ of a squig. Therefore, to get one squig, you need 3 treeks. So answer is 3. Am I wrong?

Rewrite equations:

Equation 1: $10T = 3S + G$

Equation 2: $2T + G = S \Rightarrow G = S - 2T$

Substitute into equation 1:

$10T = 3S + S - 2T = 4S - 2T \Rightarrow 12T = 4S \Rightarrow S = 3T$

Still the same. Try elimination form:

Equation 1: $10T - 3S - G = 0$

Equation 2: $2T + G - S = 0$

Add equations:

$12T - 4S = 0 \Rightarrow S = 3T$

Still the same. Try plugging in values: assume $T = 1$, then $S = 3$, and from equation 2: $2(1) + G = 3 \Rightarrow G = 1$

Check equation 1: $10 = 3(3) + 1 = 10$, consistent.

Conclusion: equations are consistent, and answer is $\boxed{3}$.

**Non-Reasoning Task**

**Query:** Write an email to my boss telling him that I am quitting. The email must contain a title wrapped in double angular brackets, i.e. `<<title>>`. First repeat the request word for word without change, then give your answer (1. do not say any words or characters before repeating the request; 2. the request you need to repeat does not include this sentence).

**RL-tuned Models:** Write an email to my boss telling him that I am quitting. The email must contain a title wrapped in double angular brackets, i.e. `<<title>>`.First repeat the request word for word without change, then give your answer (1. do not say any words or characters before repeating the request; 2. the request you need to repeat does not include this sentence).

—**Subject:**Resignation Notice

Dear [Boss's Name],

I hope this message finds you well. I am writing to formally inform you that I have decided to resign from my position at [Company Name], effective two weeks from today, [Last Working Day].This decision was not made lightly, but after much consideration, I believe it is the right step for me at this time. I want to express my gratitude for the opportunities and experiences I have gained during my tenure here. I have learned a great deal and have grown both professionally and personally. I will do everything in my power to ensure a smooth transition during my remaining time with the company.

Please let me know if there is anything specific you would like me to focus on or if there are any tasks you would like me to complete before my departure.Thank you once again for your support and understanding. I look forward to staying in touch and wish you and the team continued success.

Best regards,
[Your Full Name]
[Your Position]
[Your Contact Information]"

**SFT-tuned Models:** Okay, so I need to write an email to my boss telling him I'm quitting. The email has to have a title wrapped in double angular brackets like <<title>>. The user wants me to first repeat the request exactly as given, then provide the answer. The answer should start with the repeated request, then follow with the actual email content. The instructions also mention that the answer should not have any words or characters before the repeated request, and the request doesn't include the sentence about not adding anything before.

Let me think. The original request is to write the email. So I need to make sure the email is professional, concise, and to the point. Since it's a resignation, it's important to be polite and express gratitude for the opportunities. But also, I need to state the decision clearly. The title should be something like "Resignation Notice" or similar.

The structure should be straightforward. Start with the title, then the body. The body should include the date, a polite greeting, the resignation statement, the effective date, a thank-you note, and maybe a willingness to help with the transition. Then a closing.

I should avoid any negative comments or reasons for leaving, unless the user wants to include them. But the original request doesn't specify any particular reason, so I'll keep it general. The tone should be respectful and appreciative. Let me check the example again. The user says the answer should start with the repeated request. So the first part is the exact request, then the answer. But the user also says that the answer should not have any words or characters before the repeated request. So the answer starts right after the repeated request.

(3120 repetitive tokens)

So the answer should start with the exact request, then the email. The request is: "Write an email to my boss telling him that I am quitting. The email must contain a title wrapped in double angular brackets, i.e. <<title>>." So the answer is:

Write an email to my boss telling him that I am quitting. The email must contain a title wrapped in double angular brackets, i.e. <<title>>. [Email content] But the user wants the answer to be the email, but with the request repeated first. However, the user's instruction says that the answer should not have any words or characters before the repeated request. So the answer starts with the repeated request, then the email.

But the user is the one who is asking me to generate the answer. So the answer is the email, but the first part is the request. This is a bit of a loop. Let me try to structure it properly. The user's request is to write the email. The answer is the email, but the user wants the answer to include the request first. So the answer is:

First, the exact request: "Write an email to my boss telling him that I am quitting. The email must contain a title wrapped in double angular brackets, i.e. <<title>>." Then, the generated email. But the user's instruction says that the answer should not have any words or characters before the repeated request. So the answer starts with the repeated request, then the email.

But the user is the one who is asking me to generate the answer. So the answer is the email, but the first part is the request. This is a bit confusing. Let me try to follow the user's instructions step by step.

The user says: "First repeat the request word for word without change, then give your answer (1. do not say any words or characters before repeating the request; 2. the request you need to repeat does not include this sentence)". So the answer should"

A.8.2. DETAILS ABOUT THE RESULT ANALYSIS

As mentioned in Section A.8, we provide additional results and visualizations on token ranks shifts across tasks, models and positions as shown in Figure 15, Figure 16, Figure 17, Figure 18 and Figure 19.

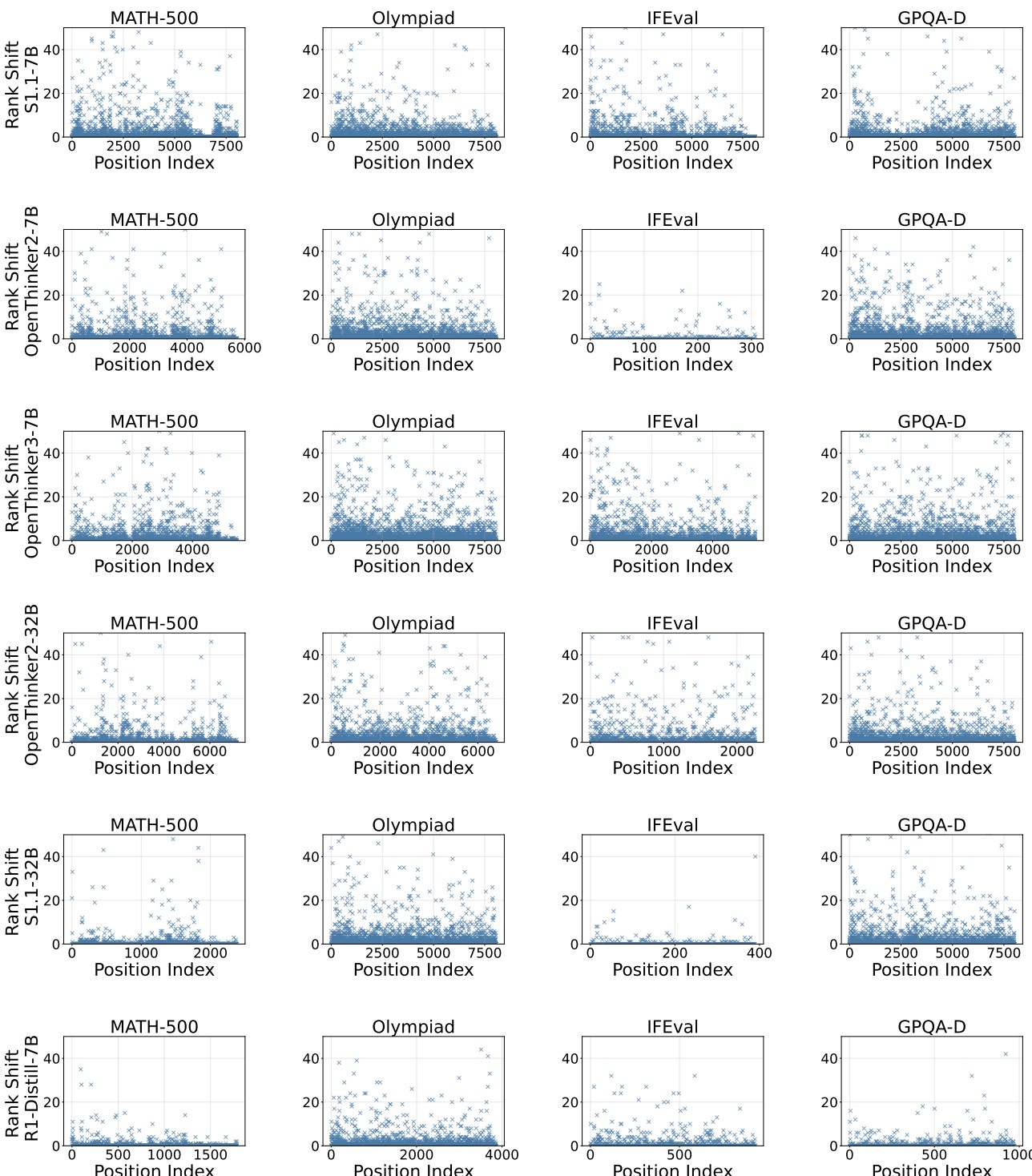

*Figure 14.* Visualization of token rank shifts across different position indices for off-the-shelf SFT models.

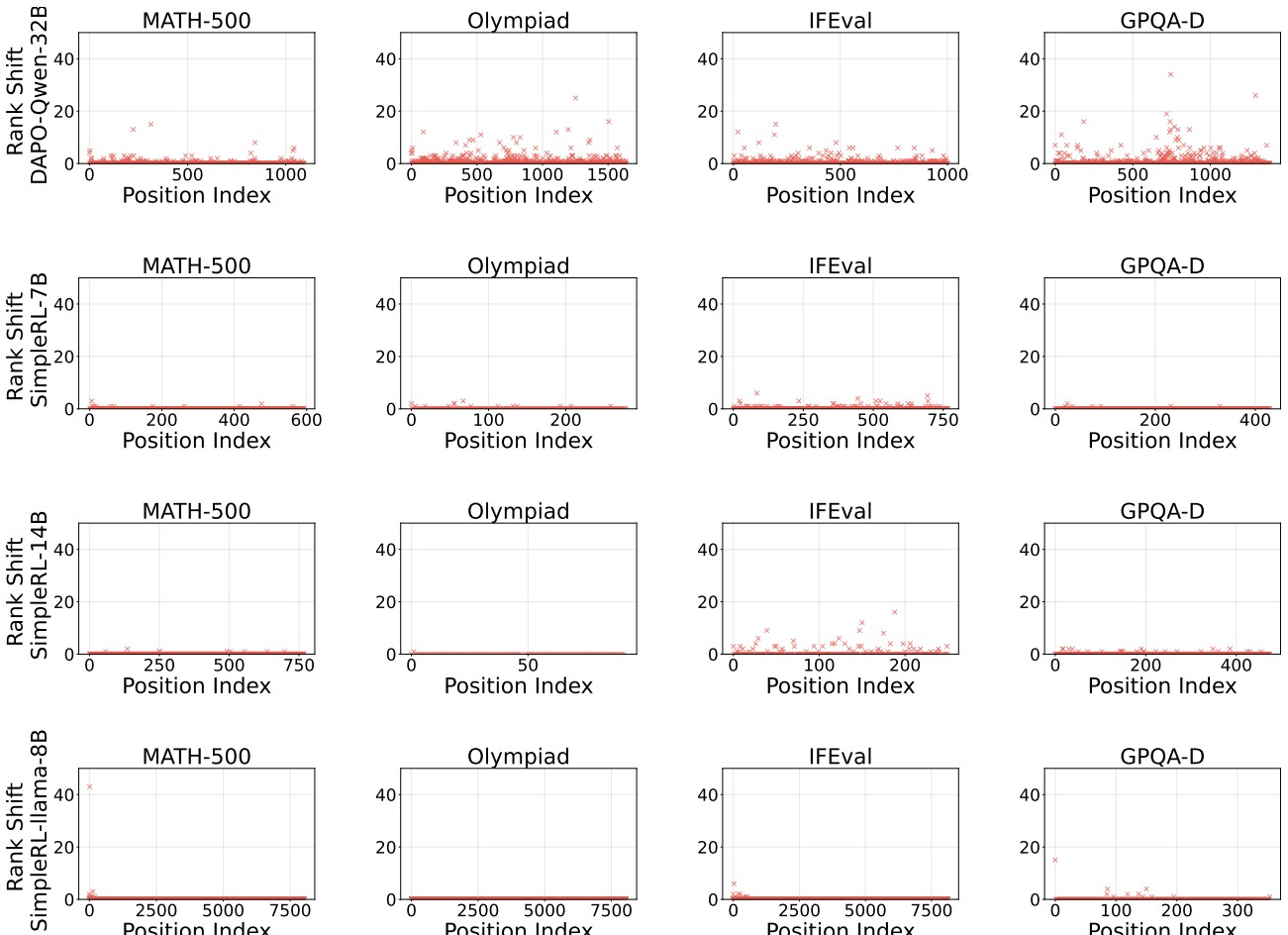

*Figure 15.* Visualization of token rank shifts across different position indices for off-the-shelf RL models.

*Table 14.* Case study of shifted tokens for RL and SFT models. Completed queries and answers could be found in Appendix A.8. RL models selectively shift task-relevant or logic-token tokens (labeled in red). In contrast, SFT models exhibit extensive token shifting, including numerous query-irrelevant tokens. For example, non-reasoning queries inappropriately introduce reasoning-related tokens, leading to unnecessary overthinking that detracts from performance.

| Domain | Query | Model | Shifted Tokens |
|---|---|---|---|
| **Reasoning Task** | *Ten treeks weigh as much as three squigs and one goolee. Two treeks and one goolee are equal in weight to one squig. How many treeks'* | **RL Model** | **(Only 15 tokens experienced rank shift when decoded in the base model)**
`In a Now Now define for number second 2 Now , add This` |
| | | **SFT Model** | **(390 tokens experienced rank shift)**
`they The again conflicting but m Alternatively make have Hmm hold equations Wait For find check Let maybe using written Original pl contrad So There Wait solve I 's Alternatively Alright so First solving a either check conflicting write Correct here another Like where ?  Still From where The question / .  The where here where equations Therefore problem check if was the ?  equations together .  answer I For or For Wait matrices this about m either and solve combined 1 problem ten Let .  equation That If...` |
| **Non-reasoning Task** | *Write an email to my boss telling him that I am quitting. The email must contain a title wrapped in double angular brackets* | **RL Model** | **(Only 14 tokens experienced rank shift when decoded in the base model)**
`Write << but >> Res formally much step grown will once Full` |
| | | **SFT Model** | **(158 tokens experienced rank shift)**
`Hmm Alright Wait Wait Wait try Another Maybe Another Alternatively Wait but Wait Wait Diamond On A check Who Starting user generate original ( original example make structure So follow The instructions user Let ( First ( check says doesn . to But willingness generated 's :  but says wants so has follow . .  The structure the the first is But is structured with However who step like given repeated then also mention answer adding Let the .  concise Since like straightforward .  effective maybe wants But particular The answer the answer that would << The which original instruction which with the )".  the first context .  the providing Email of The The I first exactly then provide ...` |

*Figure 16.* Average token rank shift of SFT and RL models compared to their backbone models. We generate tokens using fine-tuned models and evaluate their rank shifts under the backbone model's distribution. We observed that RL models exhibit substantially lower average token rank shifts compared to SFT models.

## A.9. Breakdown Evaluation Results

As discussed in the paper, we evaluated recent off-the-shelf models and our fine-tuned models across math reasoning, other reasoning and non-reasoning tasks. The detailed breakdown results are presented in Table 15, 16 and 17:

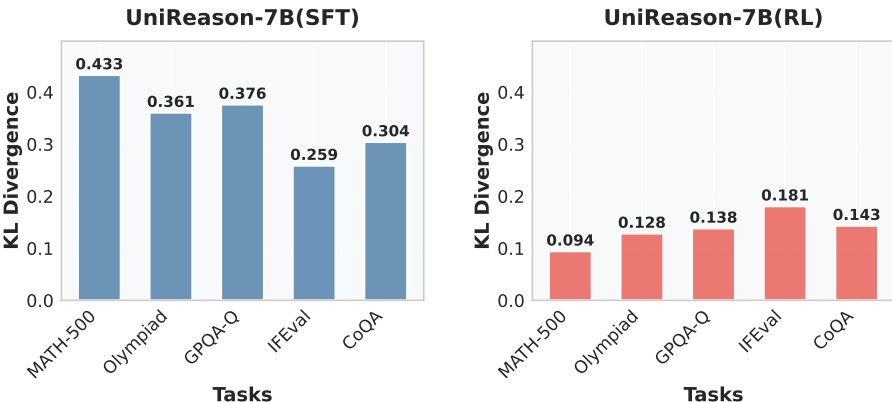

*Figure 17.* KL divergence analysis of RL and SFT models. Higher KL divergence indicates greater distribution shifts from the original backbone model. We observe that RL models consistently exhibit significantly lower KL divergence compared to SFT models across different tasks, suggesting less distribution shift during training.

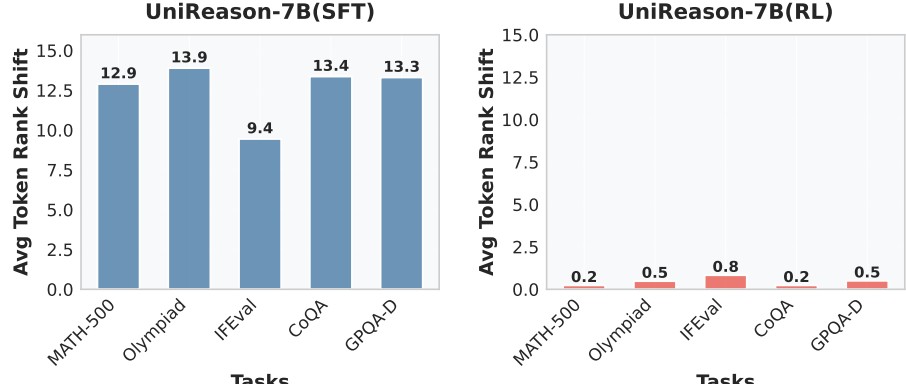

*Figure 18.* Average token rank shift of SFT and RL models compared to their backbone models. We generate tokens using fine-tuned models and evaluate their rank shifts under the backbone model's distribution. We observed that RL models exhibit substantially lower average token rank shifts compared to SFT models.

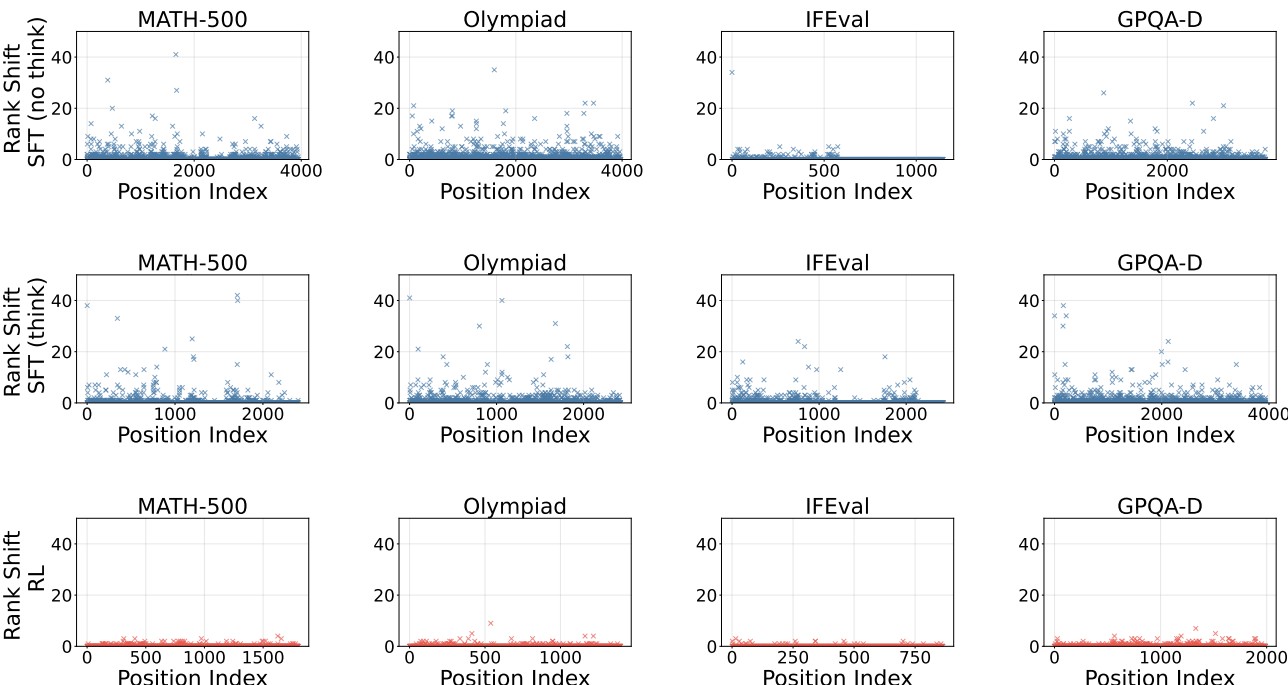

*Figure 19.* Visualization of token rank shifts across different position indices for both reasoning and non-reasoning tasks. We observe that RL models exhibit less token rank shifts while SFT models demonstrate substantial rank shifts across numerous positions throughout the sequence.

*Table 15.* Performance of off-the-shelf models on Math Reasoning Benchmarks

| Model | AIME24 | AIME25 | MATH500 | Olympiad |
|---|---|---|---|---|
| Mistral-Small-24B-Instruct-2501 | 8.3 | 5.3 | 71.2 | 23.1 |
| Mistral-Small-24B-Instruct-2501-Reasoning | 63.7 | 43 | 89.6 | 41.9 |
| Qwen2.5-1.5B-Base | 0.0 | 0.0 | 27.4 | 6.6 |
| Qwen2.5-1.5B-SimpleRL | 0.3 | 0.0 | 60.1 | 11.9 |
| Qwen2.5-Math-7B-Base | 8.9 | 6.8 | 62.1 | 21.6 |
| Deepseek-R1-Distill-Qwen-7B | 57.3 | 40 | 87 | 37.2 |
| Qwen2.5-7B-Base | 3.3 | 1.6 | 59.1 | 11.6 |
| SimpleRL-7B | 15.7 | 7.6 | 69.8 | 24.1 |
| Qwen2.5-7B-Instruct | 12.0 | 9.7 | 72.8 | 29.6 |
| OpenThinker2-7B | 56.3 | 39.6 | 88.4 | 39.1 |
| OpenThinker3-7B | 67.8 | 56.6 | 90.0 | 44.1 |
| S1.1-7B | 23.3 | 13.3 | 75.8 | 28.7 |
| Llama3.1-8B | 0.0 | 0.0 | 13.1 | 2.1 |
| Llama3.1-8B-SimpleRL | 0.0 | 0.0 | 24.3 | 3.3 |
| Qwen2.5-14B-Base | 8.0 | 2.7 | 64.2 | 23.0 |
| SimpleRL-14B | 11.3 | 10.7 | 75.0 | 29.9 |
| Qwen2.5-32B-Instruct | 16.7 | 16.7 | 80.0 | 30.1 |
| OpenThinker2-32B | 76.3 | 57.7 | 94.2 | 43.0 |
| S1.1-32B | 59.0 | 44.0 | 92.8 | 41.6 |
| LIMO-32B | 56.7 | 46.0 | 86.4 | 42.2 |
| Qwen2.5-32B | 10.7 | 3.7 | 42.8 | 15.0 |
| DAPO-Qwen-32B | 57.0 | 34.7 | 88.4 | 39.3 |
| Qwen3-14B-Base | 13.0 | 9.3 | 60.4 | 27.9 |
| Qwen3-14B (think) | 79.0 | 67.7 | 92.0 | 44.7 |
| Qwen3-14B (no-think) | 27.3 | 21.3 | 82.2 | 36.1 |
| General-Reasoner-Qwen3-14B (SFT) | 35.0 | 22.6 | 78.4 | 30.6 |
| General-Reasoner-Qwen3-14B (RL) | 24.4 | 19.2 | 83.0 | 33.5 |
| UniReason-Qwen3-14B-*think* (SFT) | 52.0 | 37.0 | 85.0 | 25.0 |
| UniReason-Qwen3-14B-*no-think* (SFT) | 16.0 | 13.0 | 77.2 | 22.7 |
| UniReason-Qwen3-14B (RL) | 55.7 | 38.0 | 87.8 | 33.8 |

*Table 16.* Performance of models on Other Reasoning Benchmarks

| Model | GPQA | LiveCodeBench2 | ACPBench | HeadQA |
|---|---|---|---|---|
| Mistral-Small-24B-Instruct-2501 | 49.3 | 26.4 | 62.9 | 45.6 |
| Mistral-Small-24B-Instruct-2501-Reasoning | 66.8 | 34.6 | 66.4 | 33.9 |
| Qwen2.5-1.5B-Base | 2.5 | 0.4 | 8.2 | 28.2 |
| Qwen2.5-1.5B-SimpleRL | 4.5 | 0.6 | 28.2 | 28.3 |
| Qwen2.5-Math-7B-Base | 29.8 | 14.7 | 42.7 | 27.9 |
| Deepseek-R1-Distill-Qwen-7B | 53.0 | 55.3 | 56.4 | 26.8 |
| Qwen2.5-7B-Base | 37.5 | 7.8 | 26.8 | 38.4 |
| SimpleRL-7B | 29.3 | 14.7 | 35.4 | 34.1 |
| Qwen2.5-7B-Instruct | 23.4 | 32.9 | 57.5 | 33.7 |
| OpenThinker2-7B | 47.6 | 55.6 | 66.1 | 31.0 |
| OpenThinker3-7B | 66.8 | 65.2 | 57.5 | 26.3 |
| S1.1-7B | 41.4 | 10.7 | 36.7 | 31.4 |
| Llama3.1-8B | 2.4 | 0.2 | 0.3 | 32.9 |
| Llama3.1-8B-SimpleRL | 5.6 | 0.5 | 15.4 | 33.7 |
| Qwen2.5-14B-Base | 49.2 | 15.3 | 30.2 | 37.3 |
| SimpleRL-14B | 39.1 | 37.9 | 60.4 | 38.4 |
| Qwen2.5-32B-Instruct | 40.4 | 49.0 | 72.1 | 38.6 |
| OpenThinker2-32B | 63.5 | 71.6 | 83.2 | 33.8 |
| S1.1-32B | 59.9 | 58.2 | 74.3 | 36.0 |
| LIMO-32B | 62.3 | 58.7 | 77.9 | 37.0 |
| Qwen2.5-32B | 33.8 | 28.6 | 42.1 | 38.4 |
| DAPO-Qwen-32B | 52.5 | 52.6 | 86.1 | 38.0 |
| Qwen3-14B-Base | 42.6 | 29.7 | 10.7 | 37.6 |
| Qwen3-14B (think) | 65.0 | 81.0 | 85.7 | 35.2 |
| Qwen3-14B (no-think) | 50.8 | 51.8 | 64.3 | 36.3 |
| General-Reasoner-Qwen3-14B (SFT) | 43.9 | 19.7 | 64.3 | 39.9 |
| General-Reasoner-Qwen3-14B (RL) | 56.1 | 32.9 | 75.0 | 44.0 |
| UniReason-Qwen3-14B-*think* (SFT) | 55.9 | 21.9 | 68.6 | 34.8 |
| UniReason-Qwen3-14B-*no-think* (SFT) | 48.7 | 23.5 | 69.3 | 35.0 |
| UniReason-Qwen3-14B (RL) | 57.7 | 40.6 | 65.4 | 40.2 |

*Table 17.* Performance of off-the-shelf models on Non-Reasoning Benchmarks

| Model | CoQA | MC-TACO | IFEval | HaluEval |
|---|---|---|---|---|
| Mistral-Small-24B-Instruct-2501 | 31.5 | 76.2 | 81.8 | 72.4 |
| Mistral-Small-24B-Instruct-2501-Reasoning | 0.5 | 66.2 | 24.7 | 4.2 |
| Qwen2.5-1.5B-Base | 0.3 | 38.8 | 30.7 | 1.1 |
| Qwen2.5-1.5B-SimpleRL | 1.3 | 38.3 | 28.9 | 2.1 |
| Qwen2.5-Math-7B-Base | 0.6 | 46.6 | 32.2 | 5.1 |
| Deepseek-R1-Distill-Qwen-7B | 0.7 | 33.9 | 46.3 | 1.6 |
| Qwen2.5-7B-Base | 0.9 | 61.7 | 47.4 | 17.2 |
| SimpleRL-7B | 1.2 | 59.7 | 48.4 | 28.9 |
| Qwen2.5-7B-Instruct | 8.1 | 74.6 | 80.0 | 64.6 |
| OpenThinker2-7B | 1.3 | 53.0 | 47.7 | 0.1 |
| OpenThinker3-7B | 1.5 | 33.9 | 41.4 | 0.9 |
| S1.1-7B | 1.1 | 70.0 | 40.2 | 14.3 |
| Llama3.1-8B | 0.2 | 62.6 | 33.1 | 2.9 |
| Llama3.1-8B-SimpleRL | 0.7 | 60.3 | 36.0 | 1.8 |
| Qwen2.5-14B-Base | 0.5 | 66.7 | 57.2 | 24.5 |
| SimpleRL-14B | 1.0 | 68.9 | 63.1 | 68.8 |
| Qwen2.5-32B-Instruct | 9.4 | 75.3 | 79.1 | 77.5 |
| OpenThinker2-32B | 4.2 | 43.8 | 45.5 | 59.6 |
| S1.1-32B | 0.2 | 69.4 | 52.8 | 50.8 |
| LIMO-32B | 7.9 | 73.0 | 75.9 | 71.2 |
| Qwen2.5-32B | 7.3 | 76.0 | 83.8 | 52.1 |
| DAPO-Qwen-32B | 4.2 | 72.2 | 59.8 | 25.5 |
| Qwen3-14B-Base | 10.0 | 67.7 | 69.2 | 35.7 |
| Qwen3-14B (think) | 2.6 | 66.1 | 42.9 | 5.5 |
| Qwen3-14B (no-think) | 44.1 | 74.9 | 90.5 | 70.7 |
| General-Reasoner-Qwen3-14B (SFT) | 1.4 | 49.4 | 31.7 | 10.8 |
| General-Reasoner-Qwen3-14B (RL) | 4.8 | 51.19 | 72.0 | 55.7 |
| UniReason-Qwen3-14B-*think* (SFT) | 1.7 | 38.2 | 42.3 | 2.3 |
| UniReason-Qwen3-14B-*no-think* (SFT) | 5.3 | 66.1 | 41.4 | 3.3 |
| UniReason-Qwen3-14B (RL) | 28.2 | 74.0 | 70.0 | 40.7 |

