# OpenReview forum: "Does Math Reasoning Improve General LLM Capabilities? Understanding Transferability of LLM Reasoning"
_ICML.cc/2026/Conference — ICML 2026 regular_

### Official Review · Reviewer_MZFE · 2026-02-22

**Soundness:** 3
**Presentation:** 4
**Significance:** 3
**Originality:** 2
**Overall Recommendation:** 4
**Confidence:** 3

**Summary:**

This paper conduct empirical study on exploring whether math reasoning improvement can generalize to other domains via different training paradigms (off-policy/on-policy SFT/RL). Through comprehensive experiments, the authors conclude that on-policy learning is the key factor that affects the generalizability. The authors also make detailed analysis on understanding the mechanism behind the transferability of on-policy RL, including PCA analysis, token rank analysis, etc. This paper provides a thorough analysis on the transferability of math reasoning.

**Compliance With Llm Reviewing Policy:**

Affirmed.

**Final Justification:**

My questions are addressed. I keep my scores.

**Key Questions For Authors:**

Please see the weakness part.

**Limitations:**

Yes.

**Strengths And Weaknesses:**

## Strengths

(1) The paper is well-written, the structure is clear.

(2) The evaluation is comprehensive. The authors conduct thorough experiments to analyze the RL/SFT transferability of math reasoning. The included tasks, models, and methods are broad.

(3) The deep analysis made in Section 3, 4 and 5 is good and provide some interesting insights.

## Weaknesses

(1) The major concern is about the novelty and the difference from previous works [1,2,3]. The RL transferability has also been analyzed in these works.

(2) Main experiments are conducted on Qwen3 models. More results on other backbones can strengthen the paper.

---
[1] Chu, Tianzhe, et al. "SFT Memorizes, RL Generalizes: A Comparative Study of Foundation Model Post-training." ICML 2025

[2] Hu, Chuxuan, et al. "Breaking Barriers: Do Reinforcement Post Training Gains Transfer To Unseen Domains?." arxiv 2025

[3] Zhou, Ruochen, et al. "Does Learning Mathematical Problem-Solving Generalize to Broader Reasoning?." arxiv 2025

---

> ### Author Rebuttal · Authors · 2026-03-31
>
> We thank the reviewer for the positive assessment of writing quality and comprehensiveness, and for the specific concerns about novelty and backbone diversity.
>
> ### W1: On novelty relative to prior works [1], [2], [3]
>
> We carefully distinguish our contributions from each cited work:
>
> **[1] Chu et al. (2025)**: This paper studies rule-based and navigation environments, and compares SFT vs RL on in-environment OOD, where their definition of OOD is a change in rules. Our study is focused on reasoning models and is on a new question of whether reasoning tuning will change the model's original general capability and improve other unseen reasoning tasks as well. And find that on-policy natural RL is crucial for transferability. The research question, setting, and findings are different between the two papers.Specifically, our work differs in: (a) we evaluate **20+ off-the-shelf models** with a unified quantitative metric (TI) rather than a small controlled study; (b) we **decompose RL's advantage** into four factors (sampling distribution, credit assignment, KL regularization, negative gradients) via a controlled ablation (Table 4); (c) we show that **on-policy SFT achieves nearly the same transferability as on-policy RL**, which directly challenges the framing of "SFT memorizes, RL generalizes" by showing the key variable is the sampling strategy, not the algorithmic family; (d) we provide mechanistic evidence via PCA shift and token-rank analyses.
>
> **[2] Hu et al. (2025)**: Both our paper and Hu et al. (2025) investigate the critical problem of reasoning transferability and reach a shared empirical consensus: RL post-training gains do not universally transfer to all unseen domains, especially those with different reasoning patterns. However, our work fundamentally differs in depth, focus, and methodology in the following ways:
> -  "Why" vs. "What": our work dives into the mechanistic reasons behind transferability. Through our latent space PCA and token distribution KL-divergence analyses (Sections 3 & 4), we empirically demonstrate how the feature geometry collapses under SFT for non-reasoning inputs, and why RL preserves the base model's internal diversity.
> - SFT vs. RL and Component Decomposition: Hu et al. primarily focus on the boundaries of RLVR across domains. In contrast, our core narrative contrasts SFT with RL. We go a step further by actively decomposing the RL pipeline (Section 5) into specific components (on-policy vs. off-policy sampling, advantage weighting, negative gradients) to isolate the exact causal mechanism. We conclude that on-policy generation is the fundamental driver of successful transfer.
>
> **[3] Zhou et al. (2025)**: Studies generalization of specific math tasks to broader reasoning like scientic reasoning, but does not analyze behaviors outside of reasoning category or include categories like agent planning, and behaviors on the on-policy/off-policy distinction or provide the mechanistic token/representation-level analyses we present.
>
> The critical novel finding absent from all three works are: 1. **on-policy SFT is nearly as transferable as on-policy RL** (Table 4), which refines the community narrative from "RL generalizes, SFT memorizes" to "on-policy training generalizes, off-policy training degrades.", 2. we conducted mechanistic analyses to show what happened within the model. We again thank reviewer for noting these high-quality related works, and we will address this distinction in the related work section and contribution section.
>
> ### W2: On backbone diversity
>
> We agree that the controlled study is primarily based on Qwen3-14B. However, we want to highlight the evidence for cross-backbone generality already in the paper:
>
> 1. **Appendix A.1 (Table 5)**: We run the full SFT vs. RL comparison on **Qwen3-4B** and **Llama3.1-8B**, both showing the same qualitative pattern. The Llama result is particularly important as it involves a different model family entirely.
> 2. **Large-scale audit (Section 2.1, Figure 2, Table 11)**: Our benchmarking, TI comparison covers models from diverse families and across different model sizes: Qwen2.5, Qwen3, Llama3.1, and Mistral.
> 3. **PCA analysis (Appendix A.6, Table 12)**: We run PCA shift analysis on Qwen2.5-7B, Qwen3-14B, Llama3.1-8B, and Qwen2.5-32B models, all showing consistent results.
> 4. **Token rank analysis (Appendix A.8, Figure 16 - 20)**: We run token rank analysis on Qwen2.5-7B, Qwen3-14B, Llama3.1-8B, Qwen2.5-32B base models and multiple models finetuned on these backbones, all showing consistent results.

---

> > ### Author Rebuttal · Reviewer_MZFE · 2026-04-03
> >
> > Thanks for the clarification, I maintain my scores.

---

### Official Review · Reviewer_TwBE · 2026-03-11

**Soundness:** 4
**Presentation:** 3
**Significance:** 3
**Originality:** 4
**Overall Recommendation:** 5
**Confidence:** 3

**Summary:**

This paper investigates whether improved math reasoning abilities in large language models transfer to other general capabilities. The authors evaluate over 20 open-weight reasoning-tuned models across math reasoning, other reasoning (scientific QA, coding, planning), and non-reasoning tasks (conversational QA, instruction following, hallucination evaluation). They propose a Transferability Index metric to quantify how well models transfer performance from mathematics to other domains. The key finding is that the fine-tuning paradigm is the dominant factor predicting transferability: RL-tuned models consistently achieve higher transferability indices on both other-reasoning and non-reasoning tasks, whereas SFT-trained models often yield negative transfer on non-reasoning benchmarks. To validate this, the authors conduct a controlled study on latent PCA shift and KL-divergence on token distributions, which reveals that SFT induces substantial representation and output drift, especially for non-reasoning inputs, while RL preserves the geometry of internal features and the stability of token distributions. An ablation study further decomposes RL's advantage, finding that sampling distribution (on-policy vs. off-policy) is the most critical factor, with credit assignment and negative gradients also contributing.

**Compliance With Llm Reviewing Policy:**

Affirmed.

**Final Justification:**

Most of my concerns/questions are addressed. Regarding W1, the authors have promised to weaken and narrow the claim as well as clarify the settings, which I think is reasonable. I thus maintain my positive score.

**Key Questions For Authors:**

1. How would the converse problem be different or similar to your research question (i.e. other reasoning/non-reasoning transferring to math reasoning)? I know it might be hard to conduct training because data curation is not as obvious as math reasoning, but some intuition or preliminary studies would be meaningful.

2. Do you have an explanation on why RL on math tasks would improve non-reasoning capabilities? I understand that SFT can hurt due to forgetting and disruptive updates, but RL on math would, in principle, not interfere with non-reasoning tasks?

**Limitations:**

Mostly. I think other training paradigms than SFT and RL could be studied since the research question is not limited to these two. For instance, [on-policy distillation](https://thinkingmachines.ai/blog/on-policy-distillation/).

**Strengths And Weaknesses:**

**Strengths:**
- The research question is well-defined, and the experiment setup is clear and sound. Both the transferability index experiment and the subsequent controlled studies are well-motivated.
- It presents two complementary angles of mechanistic analysis on why SFT generalizes poorly compared with RL, with clear visuals and textual explanation.
- The ablation on RL components made it clearer how generalization from math to other domains occurs.
- Overall, it provides rather comprehensive empirical evidence for the common belief that SFT induces memorization and RL promotes generalization.

**Weaknesses:**
1. The claim on credit assignment, saying "uniform weighting vs advantage-weighted updates that emphasize task-salient tokens", seems wrong. For GRPO with outcome rule-based reward, the advantage on each token is also uniform. From what I understand, this is not the credit assignment issue in RL that treats each token differently based on saliency. It reweights samples in a group instead of on a token level.
2. The hyperparameters do not seem sufficiently tuned. To establish sound claims, I think tuning the batch size and learning rate to be optimal in RL/SFT separately is necessary. Just by eyeballing the parameters, the batch size of 512 might be too large for SFT, which can hurt generalization. It's not so reasonable to keep RL and SFT hyperparameters the same, and rather, you might need to tune them separately.
3. Some notations are not well-defined and used. For instance, when defining transferability, $R_b^\text{model}$, $R_b^\text{base}$ should be defined as trained and original base model results on a benchmark $b$. When describing the latent PCA process, $\mathbf{H}_i$'s dimension should be clarified. The notations in Table 3 are not explained in the text either.

---

> ### Author Rebuttal · Authors · 2026-03-31
>
> We thank the reviewer for the detailed and technically precise feedback.
>
> ### W1: On the credit assignment claim for GRPO
>
> Thanks for raise a valid and important technical point. In GRPO, the advantage $A(x, y)$ is computed at the sample level instead of token level. Therefore, we agree with reviewer that the phrase “emphasize task-salient tokens” is imprecise in this context.
>
> Our intention was to highlight the contrast between sample-level selectivity in RL and the uniform treatment of all demonstrations in SFT. RL assigns higher weight to successful trajectories and lower weight to failed ones, whereas SFT applies uniform weight across all teacher  sequences regardless of their quality. This is the sense in which RL performs credit assignment at the sample level, even though within a single trajectory all tokens receive the same advantage value.
>
> We will revise the credit assignment discussion in Section 5 to be more precise:
>
> - In our framework, "credit assignment" refers to **sample-level advantage weighting**: RL uses advantage $A_t$ to upweight high-reward samples and downweight low-reward ones, whereas SFT uses uniform weight $w=1$ on all teacher demonstrations.
> - We will explicitly clarify that we do not claim token-level saliency weighting within GRPO, and we will update the description in Section 5 accordingly to avoid confusion.
>
>
> ### W2: On hyperparameter tuning
>
> The reviewer raises a legitimate concern. We note:
>
> 1. We used batch size 512 for both SFT and RL to ensure a controlled comparison where the only varying factor is the training objective (not the optimization setup). If we tuned batch size separately, it would conflate hyperparameter differences with algorithmic differences.
> 2. We ran an ablation with a different learning rates for SFT on Qwen3-4B and Llama3.1-8B (Table 5 in Appendix A.1). Reducing the learning rate uniformly hurts math performance without improving non-reasoning transfer.
> 3. We will provide additional experiment result with batch size 128 for SFT once it is done.
>
> We acknowledge that a more exhaustive hyperparameter search would further strengthen the claims. We will note this as a limitation and plan to include additional ablations.
>
> ### W3: On notation clarity
>
> Thank you for identifying these gaps. We will:
>
> 1. In the TI definition (Section 2.1): explicitly define $R^{model}_b$ as the accuracy of the fine-tuned model on benchmark $b$, and $R^{base}_b$ as the corresponding base model accuracy.
> 2. In the PCA section: clarify the dimension of hidden states $\mathbf{H}^{(*)}_i \in \mathbb{R}^{N \times d}$ where $N$ is the number of input tokens and $d$ is the hidden dimension.
> 3. In Table 3: add a caption footnote explicitly defining each column symbol and its mapping to the loss in Eq. (3).
>
> ### Q1: On the converse direction (other reasoning / non-reasoning to math)
>
> Thanks for rising a natural and interesting question. We do not study it in the current paper primarily because:
>
> 1. **Data curation**: constructing high-quality non-reasoning datasets for RL comparable to math is harder.
> 2. **Scope**: To focus on understanding difference in tuning method, we simplified our source of dataset for a cleaner comparsion. Our paper's focus is the specific question of whether math-specialized post-training generalizes, motivated by the community's heavy investment in math as a proxy task.
>
> Intuition:
> For non-reasoning domains, if on-policy training is the key driver, we would expect similar transfer behavior in the reverse direction. Which means non-reasoning tuning will not hurt model's behavior on math reasoning if we conduct on-policy post-training.
>
> For general reasoning domains, we expect the two directions to be asymmetric. Mathematical reasoning typically requires structured multi-step deductions, whereas general reasoning tasks rely more domain knowledge and are less structured. As a result, improvements from general-domain training may not directly translate into gains in math reasoning.
>
> Nevertheless, we believe this is a promising direction for future studies.
>
>
> ### Q2: On why RL on math improves non-reasoning capabilities
>
> Thanks for this excellent mechanistic question. Our PCA and KL-divergence analyses in Sections 3-4 suggest the answer is **not** that RL actively improves non-reasoning capabilities, but rather that it **preserves** the base model's non-reasoning representations. On-policy RL updates are small and targeted (low PCA shift, low KL divergence, few token-rank shifts), leaving the model's general-domain feature space largely intact. Since the base Qwen3-14B-Base already has strong non-reasoning capabilities, preserving them while improving math leads to positive TI$_{non}$.
>
> In contrast, off-policy SFT causes large representational drift even in non-reasoning layers, actively overwriting general-domain features with math-specific patterns.
>
> We greatly appreciate the reviewer’s time and their detailed, high-quality feedback.

---

> > ### Author Rebuttal · Reviewer_TwBE · 2026-04-02
> >
> > W1: Thanks for the clarification. However, I don't think you can directly compare SFT with RL on sample-level weighting. For each question, SFT only has one teacher rollout to train on, while RL is weighting many rollouts from the same question. It's hard to ablate this component between SFT and RL due to difference in optimization objective, at least I don't think your current experiment can prove that such weighting scheme in RL contributes to the generalization behavior.

---

> > > ### Author Response · Authors · 2026-04-07
> > >
> > > We thank the reviewer for this careful and important comment. We apologize that our ablation setup was not clearly explained in the original draft.
> > >
> > > The controlled ablation experiments use the same number of rollouts (n=8), including on-policy RL, on-policy SFT, off-policy RL, and off-policy SFT. In our on-policy SFT setting, we use the same on-policy rollout pipeline as in on-policy RL: the policy model generates 8 samples for each prompt, after which we perform rejection sampling and retain only the correct trajectories for update. The update on these retained trajectories is then performed with uniform weight 1 i.e., without advantage weighting.
> > >
> > > When comparing on-policy RL against on-policy SFT, as the reviewer mentioned, it is hard to isolate the effect of weighting alone. This comparison jointly changes two ingredients relative to RL: (1) advantage-based weighting is removed, and (2) unsuccessful / negative trajectories are no longer used for learning due to rejection sampling.
> > >
> > >
> > > We observe that after controlling for the sampling distribution, on-policy RL still outperforms on-policy SFT. We therefore interpret this result more cautiously as evidence that RL’s advantage is not due to on-policy sampling alone, but also depends on the additional training signal provided by reweighting and negative trajectories. However, we agree that this experiment does not fully disentangle the independent contribution of weighting from that of negative examples.
> > >
> > > In our understanding, refering to Equation 1 and Equation 2, the difference of objective between RL and SFT in our formulation comes from sampling distribution, the advantage-based weighting, and the KL regularization term. During On-policy SFT, $y^{\star}$ in Equation 1 is the answer that comes from policy model, so it is identical to y in in Equation 2 of On-policy RL. For Off-policy SFT and RL, the policy model is both Qwen3-32B. Therefore, the difference on objectives lies on sampling distribution, the advantage-based weighting, and the KL regularization term. In this experiment, we set KL to 0 for RL, and controlled sampling distribution.
> > >
> > > Our ablations were designed as control studies around these understandings. At the same time, we agree with the reviewer that the distinction between RL and SFT may be more complicated than our current experimental decomposition captures. Also, we fully acknowledge that the claim in the current draft was not stated precisely enough. In the revision, we will clarify the experimental setup, narrow the claim, and explicitly state that this ablation supports the importance of weighting and negative examples jointly.
> > >
> > > We would like to thank the reviewer again for the insightful comments!

---

### Official Review · Reviewer_ZYxD · 2026-03-15

**Soundness:** 3
**Presentation:** 2
**Significance:** 3
**Originality:** 2
**Overall Recommendation:** 4
**Confidence:** 4

**Summary:**

This paper investigates whether improvements in mathematical reasoning effectively transfer to general LLM capabilities by evaluating over 20 open-weight reasoning models across math, other reasoning tasks, and non-reasoning tasks. To quantify the transferability, it introduces the “Transferability Index", a metric measuring the ratio of performance improvement in target domains relative to the source (math) domain. Controlled experiments and ablation studies reveal that on-policy fine-tuning is the critical mechanism for successful transfer.

**Compliance With Llm Reviewing Policy:**

Affirmed.

**Key Questions For Authors:**

Please refer to the weaknesses.

**Limitations:**

yes.

**Strengths And Weaknesses:**

Strengths:
- Soundness: The introduction of the “Transferability Index” provides a quantitative way to measure the cross-domain transfer.
- Soundness: The paper conducts comprehensive experiments to investigate the cross-domain transfer problem, i.e., different architectures and sizes, detailed ablation studies, and mechanistic explanations.
- Presentation: The paper is well organized and easy to follow.
- Significance: The paper tries to tackle a highly relevant problem in the LLM community, i.e., the transferability of reasoning capabilities in LLMs.
- Originality: Although the core problem is not entirely new, the specific findings regarding the transferability of math reasoning and the specific causal factors have a touch of novelty.

Weaknesses:
- Soundness: The categorization of benchmarks into “reasoning” and “non-reasoning” could be debated. Some tasks labeled as non-reasoning might still implicitly require reasoning, potentially blurring the lines of what exactly is being “forgotten” or “transferred”.
- Presentation: Figure 3 is too small to clearly demonstrate the results. Maybe the authors could select some typical results to show and leave the others in the appendix.
- Originality: As mentioned before, some findings are general, e.g., SFT can lead to catastrophic forgetting is established knowledge in the field.

---

> ### Author Rebuttal · Authors · 2026-03-31
>
> We thank the reviewer for the positive assessment of soundness, presentation, and significance.
>
> ### W1: On the benchmark categorization ("reasoning" vs. "non-reasoning")
>
> We agree this boundary is not sharp, and we appreciate the reviewer raising it explicitly. We agree that the boundary can sometimes be blurry, as many NLP tasks require implicit reasoning. Our categorization is based on explicit task formulation: "reasoning" tasks here require multi-step logic, symbolic manipulation, or algorithmic planning (Math, Code, Agent). "Non-reasoning" tasks (CoQA, IFEval, HaluEval, MC-TACO) are characterized by tasks where the primary challenge is factual recall, instruction adherence, conversational coherence, or hallucination avoidance. Where tasks do not acquire multi-step or symbolic reasoning. While all language understanding involves some form of reasoning, the cognitive demands of these tasks are qualitatively different from OlympiadBench, LiveCodeBench or GPQA.
>
>
> We will add a clarifying paragraph in Section 2.1 acknowledging this gray zone and noting that our three-category taxonomy is a practical approximation. We will also note that the consistent pattern (RL > SFT across all three groups) holds regardless of exactly where the boundary is drawn, which strengthens the robustness of our conclusions.
>
> We thank reviewer for their careful distinction on this boundary.
>
> ### W3: On SFT catastrophic forgetting being established knowledge
>
> We agree that SFT-induced forgetting is a known phenomenon in general, our novely is demonstrating why RL (and in more general term, on-policy training) is immune to this specific type of representation collapse during reasoning-specific post-training. We believe our contribution goes meaningfully beyond this:
>
> 1. We **quantify** the phenomenon at scale across 20+ open-weight reasoning models using the Transferability Index, providing the first systematic measurement of how bad the forgetting is in the context of math-specialized post-training.
> 2. We **identify on-policy sampling** as the primary mechanistic driver via controlled ablation (Table 4), which is a more specific and actionable finding than the general SFT-vs-RL observation.
> 3. We **explain mechanistically** why forgetting occurs (latent-space PCA shift, KL divergence, token-rank analysis), going beyond empirical observation to provide diagnostic tools.
> 4. We show that **on-policy SFT nearly matches on-policy RL** in transferability, which is a surprising and novel result not established by prior work.
>
> We will restructure the Introduction and Related Work to more clearly state what is new versus what builds on prior findings.
>
> ### W2: Figure 3 readability
>
> We appologize for the issue. We will update the figure for better readability.

---

> > ### Author Rebuttal · Reviewer_ZYxD · 2026-04-05
> >
> > Thanks for the detailed clarification. My original score reflects my recommendation properly, so I choose to keep it.

---

### Official Review · Reviewer_Pyi8 · 2026-03-23

**Soundness:** 2
**Presentation:** 3
**Significance:** 2
**Originality:** 2
**Overall Recommendation:** 4
**Confidence:** 3

**Summary:**

The paper under review is asking an important question whether improvements in mathematical reasoning during LLM post-training transfer onto other capabilities in  the model? The authors evaluate > 20 models (open weights) across different taks (maths, scientific AA, agent planning, coding and instruction following). Generally, the authors find that the models "fail" to transfer the capabilities from maths to other areas.

**Compliance With Llm Reviewing Policy:**

Affirmed.

**Final Justification:**

my major concerns have been addressed

**Key Questions For Authors:**

Q1. TI formula TI_g normalises improvements in relation to maths gains (TI_g=DI_g / DI_math×100). How robust is this formula to small values of the denominator?  Are teh models in the >20 survey, especially where the maths gains are small, making TI unstable?

Q2. Regarding Table 4 showing on-policy SFT scores TI_non=+30.2 and on-policy RL TI_non=32.4 (a very small gap), do authors think teh on-policy/off-policy angle is the primary finding, with RL being better largely because it is fundamentally on-policy? Could you expand on this?

Q3. Cheng at al. (2025) "Revisiting reinforcement learning for llm reasoning from a cross-domain perspective" and Hu at al. (2025) "Breaking barriers: Do reinforcement post training gains transfer to unseen domains?" seem to be very related to this work. Could you explain clearly what are the key differences both empirical and conceptual?

**Limitations:**

yes

**Strengths And Weaknesses:**

Overall the paper seems technically sound. A fair evaluation of more than 20 reasoning models on various benchmarks (sec. 2) seems solid. The ablation studies (sec. 5) aiming to separate four different components that drive RL transferability seems coherent. The proposed Transferability Index (TI) seems to be a key here as since as a metric it measures "how reasoning models can transfer their capabilities from one domain to the other". It involves several design choices including normalization, signed square-root compression, difficulty weighting, ratio normalization in relation to the math group. Understanding the sensitivity of those design choices is fundamental here. In Appendix 2 the authors include the ablations, showing the qual ordering seems robust to removing the square root transform, however the particular combination of choices is not well motivated. Specifically, teh difficulty weighing \hat w_b (page 3) and the signed square root seem ad hoc and the sensitivity to the specific selection is not sufficiently explored, which an important issue here.

Overall the paper is well presented, logically organised and in general easy to follow. Fig 3 a somewhat hard to read,  the graphs are to cluttered and the text (especially within boxes) is too small to read comfortably. It would be nice to be able to read the caption and have a better sense of what you want to highlight from this; in other words to make is as self-contained as possible. Again, the key section on the design of TI (sec 2.1) is  too brief and dense given the importance of it. It would be helpful to add additional intuition and justification for the choices made.

The paper definitely addresses an important question, which is relevant to the LLM post-training community. Selective benchmarking can be very problematic to have a more realistic picture. The question of whether on-policy training is the principla driver of transferability is a useful and actionable insight.

---

> ### Author Rebuttal · Authors · 2026-03-31
>
> We thank the reviewer for the detailed review and highlighting the importance of the problem.
>
> ### The design choices of TI
>
> We acknowledge the motivation for each design choice deserves more explicit justification and will expand Section 2.1 accordingly. Briefly:
>
> - **Difficulty weighting $\hat{w}_b$**: We upweight harder benchmarks because improvements on them are more meaningful signals of genuine capability transfer.
> - **Signed square-root compression**: The raw DI values can be dominated by a few benchmarks. The signed square-root reduces sensitivity to outliers while preserving the direction of transfer.
>
> Additionally, we provide a full design space ablation across all 8 combinations of three design choices: sqrt compression ($s_b =\operatorname{sign}(\delta_b)|\delta_b|^{1/2}$ vs. linear $s_b = \delta_b$), difficulty weighting ( $(100 - R_b^{base})$vs. uniform), and z-score normalization ($\delta_b = \Delta R_b / \sigma_g$ vs. raw $\Delta R_b$). The RL$-$SFT separation ($TI_{other}$ and $TI_{non}$ of RL models minus SFT models) is **strictly positive across all 8 combinations**, confirming that model rankings are essentially unchanged regardless of design choices. Notably, removing sqrt compression yields *larger* separations, confirming our compression is conservative. We will add Table R1 to Appendix A.2.
>
> **Table R1.** Design space ablation (* = paper's baseline).
> Sep = RL\_mean $-$ SFT\_mean; $\rho$ = Spearman vs. baseline.
>
> | Config | $Sep_{o}$ | $\rho$ | $Sep_{n}$ | $\rho$ |
> |--------|---------------:|-------:|---------------:|-------:|
> | linear, uniform, raw | 52.6 | 0.88 | 178.3 | 0.99 |
> | linear, uniform, $/\sigma_g$ | 46.3 | 0.88 | 141.3 | 0.99 |
> | linear, diff-wt, raw | 62.3 | 0.94 | 146.5 | 0.99 |
> | linear, diff-wt, $/\sigma_g$ | 54.9 | 0.94 | 116.1 | 0.99 |
> | sqrt, uniform, raw | 38.9 | 0.96 | 139.9 | 1.00 |
> | sqrt, uniform, $/\sigma_g$ | 36.6 | 0.96 | 124.5 | 1.00 |
> | sqrt, diff-wt, raw | 45.1 | 1.00 | 126.5 | 1.00 |
> | sqrt, diff-wt, $/\sigma_g$* | 42.4 | 1.00 | 112.6 | 1.00 |
>
> ### Q1: Robustness of TI when math gains (denominator) are small
>
> While instability is a valid concern, it does not arise in our paper. We report $DI_{math}$ for off-the-shelf models below:
> SimpleRL-Llama3.1-8B (0.26), UniReason-SFT-nothink (0.28), S1.1-7B (0.40), SimpleRL-1.5B (0.46), SimpleRL-14B (0.62), General-Reasoner-SFT-14B (0.63), SimpleRL-7B (0.78), General-Reasoner-RL-14B (0.78), UniReason-SFT-think (0.93), LIMO-32B (1.22), UniReason-RL-14B (1.23), S1.1-32B (1.24), OpenThinker2-7B (1.26), Deepseek-R1-Distill-7B (1.37), OpenThinker2-32B (1.45), Mistral-Reason-24B (1.45), OpenThinker3-7B (1.46), DAPO-32B (1.47).
>
> No model has $DI_{math}\leq 0$ or extremely small (min=0.255, median=1.079). We will note in Section 2.1 that TI is measured on the domain that is specifically finetuned on. A successful finetuning implies the gain on the finetuning domain(math) is not trival, which is the prior condition for our experiment. This ensures that it could transfer its capability to other domains.
>
> ### Q2: Is on-policy training the primary driver?
>
> This is precisely our main conclusion, and we appreciate the reviewer's framing. Table 4 shows that the gap between on-policy SFT and on-policy RL is indeed small, while the gap between on-policy and off-policy methods is large and consistent. This confirms that **on-policy sampling is the dominant mechanism**, and RL's additional benefits (credit assignment, negative gradients) provide some improvement on top. Our work would like to refine the community's understanding of RL: the enhanced generalization often equated with RL should be more precisely credited to on-policy training dynamics.
>
> ### Q3: Relationship to Cheng et al. (2025) and Hu et al. (2025)
>
> We thank the reviewer for flagging these concurrent works:
>
> - **Cheng et al. (2025)** study cross-domain RL generalization with a focus on reward model generalization and domain-specific behavior. They do not systematically disentangle on-policy vs. off-policy sampling as a standalone variable, nor do they propose the Transferability Index metric or conduct the latent-space PCA / token-distribution diagnostic analyses we present.
> - **Hu et al. (2025)**  Our work fundamentally differs in depth, focus, and methodology: 1. "What" vs. "Why": our work dives into the mechanistic reasons behind transferability through our PCA and token analyses (Sections 3 & 4), we demonstrate how the feature geometry collapses, and why RL preserves the base model's internal diversity. 2. SFT vs. RL & Component Decomposition: We go further than Hu et al. by actively decomposing the RL pipeline (Section 5) into specific components (on-policy vs. off-policy sampling, advantage weighting, negative gradients) to isolate the exact causal mechanism. We conclude that on-policy generation is the fundamental driver of successful transfer.
>
> ### Figure 3 readability
>
> We appologize for the issue. We will update the figure for better readability.

---

> > ### Author Rebuttal · Reviewer_Pyi8 · 2026-04-04
> >
> > Thank you for the detailed response. My major concerns have been addressed. I'm happy to raise my score

---

> > > ### Author Response · Authors · 2026-04-07
> > >
> > > Thank you for your thoughtful engagement throughout the review process! We are glad our rebuttal addressed your concerns, and we truly appreciate your willingness to raise the score.

---

### Decision · Program_Chairs · 2026-04-30

**Decision:**

Accept (regular)

**Comment:**

This paper studies an important question: whether improvements in mathematical reasoning transfer to general LLM capabilities. The empirical study is comprehensive and technically solid, and the results consistently show limited transfer across domains.
A key contribution is the identification of on-policy training as the primary driver of transferability, refining the common RL vs. SFT narrative. This provides a useful and actionable insight for the community.
The main limitations are moderate novelty and some lack of rigor in isolating causal factors, although the rebuttal addresses most concerns and does not affect the main conclusions.
Overall, this is a solid empirical paper with meaningful (though incremental) insights. I recommend accept.